*The Company of*
**Biologists**

# SOX2 and NR2F1 coordinate the gene expression program of the early postnatal visual thalamus

Linda Serra[1,*], Anna Nordin[2,3,4,*], Mattias Jonasson[2,3], Carolina Marenco[1], Guido Rovelli[1], Annika Diebels[1], Francesca Gullo[1], Sergio Ottolenghi[1], Federico Zambelli[5], Michèle Studer[6], Giulio Pavesi[5], Claudio Cantù[2,3,4,‡], Silvia K. Nicolis[1,‡] and Sara Mercurio[1,‡]

## ABSTRACT

The thalamic dorsolateral geniculate nucleus (dLGN) receives visual input from the retina via the optic nerve, and projects to the cortical visual area, where eye-derived signals are elaborated. The transcription factors SOX2 and NR2F1 are directly involved in the differentiation of dLGN neurons, based on mouse work and patient mutations leading to vision defects. However, whether they regulate each other, or control common targets is still unclear. By RNA-seq analysis of neonatal dLGN from thalamo-specific *Sox2* and *Nr2f1* mouse mutants, we found a striking overlap of deregulated genes. Among them, *Vgf*, encoding a cytokine transported along thalamic-cortical axons is strongly downregulated in both mutants. Direct SOX2 binding to some of these genes was confirmed by CUT&RUN, which identified a SOX2 chromatin-binding pattern characteristic of the dLGN. Collectively, our genetic and molecular analyses on the SOX2 and NR2F1-coregulated genes contribute to our understanding of the gene regulatory network driving the differentiation and connectivity of thalamic neurons, and the vision impairments caused by mutations in these genes.

**KEY WORDS: Sox2, Nr2f1, Vision, Thalamus, DLGN, CUT&RUN, Vgf**

## INTRODUCTION

Inherited diseases affecting vision importantly contribute to significant disability in human populations. Most frequently, the retina and the eye are involved, but other parts of the visual axis, such as thalamic nuclei and visual cortical areas can also be affected (Williamson and FitzPatrick, 2014; Cardozo et al., 2023; Graw, 2003). A wide variety of genes are known to be mutated in such diseases, ranging from transcription factors to cell membrane receptors, to secreted signaling molecules, to second messengers in signal transduction, etc. (Williamson and FitzPatrick, 2014; Cardozo et al., 2023). Interestingly, transcription factors expressed during early development with pleiotropic functions are found to be mutated in specific cases, which raises the issue of which genes among their targets are functionally important to cause the pathological phenotype. Studies in humans and mice show that mutations of two genes, *NR2F1* and *SOX2* cause important vision impairments (Williamson and FitzPatrick, 2014; Schaaf et al., 1993; Williamson et al., 1993; Mercurio, 2023). *NR2F1* mutations are responsible for the Bosch-Boonstra-Schaaf optic atrophy syndrome (OMIM #615722; BBSOAS), whereas *SOX2* mutations cause micro- or anophtalmia (OMIM #206900, Microphtalmia, syndromic 3; optic nerve hypoplasia and abnormalities of the central nervous system). Both *Sox2* and *Nr2f1* are highly expressed in the developing sensory thalamus from early developmental stages (Chou et al., 2013; Mercurio et al., 2019a). Thalamo-specific conditional knockout (cKO) mouse mutants show that inactivation of *Sox2* or *Nr2f1* affects, albeit at different degrees of severity, proper differentiation of the thalamic dorsolateral geniculate nucleus (dLGN) and the visual cortex, also causing alterations of the topographic connections relaying information to and from these functional locations (Chou et al., 2013; Mercurio et al., 2019a,b; Armentano et al., 2007; Pevny and Nicolis, 2010) The dLGN nucleus is crucial for conveying information from the retina to the primary visual cortex. Mouse mutants, conditionally deleted in the dLGN for either *Sox2* or *Nr2f1* using a RORα-Cre transgene, result in hypomorphic visual thalamus postnatally, reduced thalamic projections to the primary visual cortex (Chou et al., 2013; Mercurio et al., 2019a), significant abnormalities of the visual cortex, and important alterations of retino-thalamic connections (Mercurio et al., 2019a). We reasoned that dysregulation of a common set of target genes of NR2F1 and SOX2 might be the underlying factor causing phenotypic similarity following their thalamic (dLGN) inactivation. If this is correct, it might point to a gene set universally shared during development of the dLGN and the visual axis, as well as common to the visual phenotype of *Nr2f1* and *Sox2* mutants.

In the present work, we obtained RNA-seq data from wild type as well as *Nr2f1* or *Sox2* dLGN mutants and identified over 500 dysregulated genes in common. We show that most of the top downregulated genes in *Nr2f1* mutants are also significantly downregulated in *Sox2* mutants (44 out of 50); likewise, the large majority of the top downregulated genes in *Sox2* mutants are significantly downregulated also in *Nr2f1* mutants. These results suggest that many of the downregulated genes identified in this study contribute to at least some of the phenotypic alterations observed in mouse mutants. Functional enrichment analysis showed that deregulated genes are highly enriched in differentiated neuronal functions (axon guidance molecules, synaptic proteins, etc.). Deconvolution analysis of RNA-seq data, based on the

[1]Department of Biotechnology and Biosciences, University of Milano-Bicocca, Piazza della Scienza 2, 20126 Milano, Italy. [2]Wallenberg Centre for Molecular Medicine, Linköping University, SE-581 83 Linköping, Sweden. [3]Department of Biomedical and Clinical Sciences, Division of Molecular Medicine and Virology, Faculty of Medicine and Health Sciences, Linköping University, SE-581 83 Linköping, Sweden. [4]Science for Life Laboratory (SciLifeLab), Linköping University, SE-581 83 Linköping, Sweden. [5]Department of Biosciences, University of Milano, 20133 Milano, Italy. [6]Université Côte d'Azur, CNRS, Inserm, Institute of Biology Valrose (iBV), 06108 Nice, France.
*Joint first authors

‡Authors for correspondence (sara.mercurio@unimib.it; silvia.nicolis@unimib.it; claudio.cantu@liu.se)

M.J., 0000-0001-5528-3405; M.S., 0000-0001-7105-2957; C.C., 0000-0003-1547-5415; S.K.N., 0000-0003-0378-3808; S.M., 0000-0003-3469-8323

Biology Open

comparison with single-cell RNA-seq data previously obtained on wild-type visual thalamus (Kalish et al., 2018), revealed a substantial reduction in cells with a transcriptional identity of glutamatergic neurons already at early developmental stages, preceding phenotypic defects. Moreover, we employed the Cleavage Under Targets and Release Using Nuclease (CUT&RUN; Skene and Henikoff, 2017) approach to identify the *in vivo* genome-wide direct SOX2 binding sites in visual thalamic nuclei. This analysis allowed us to establish: i) the first full set of SOX2 binding sites in dLGN differentiated neurons, ii) the subset of targets that are transcriptionally regulated by SOX2, and iii) those that likely depend on the interplay between SOX2 and NR2F1. Finally, we also identified a general binding consensus sequence for SOX2 together with NR1F1/NR2F1 transcription factors characteristic of the dLGN that might be used by several other transcription factors involved in the differentiation of thalamic neurons.

## RESULTS

### *Sox2* and *Nr2f1* deletion in the developing visual thalamus causes deregulation of gene expression that precedes thalamic defects

To unravel the origin of the common phenotype observed in *Sox2* and *Nr2f1* thalamo-specific mouse mutants, we first stained the dLGN at postnatal day 0 (P0) (Fig. 1A) and P8 (Fig. S1B) with anti-SOX2 and anti-NR2F1 antibodies and observed high co-expression of the two proteins in thalamic differentiated neurons (Fig. 1A; Fig. S1). Then, to identify the gene regulatory network downstream of SOX2 and NR2F1 in the visual thalamus, we performed RNA-seq experiments on *ex vivo* dissected dLGN from *Sox2* or *Nr2f1* thalamic mutants (obtained via *RORα*-Cre deletion) and their control littermates at P0, before the appearance of overt morphological impairments (Chou et al., 2013; Mercurio et al., 2019a) (Fig. 1B). To note, RORα-Cre mediated deletion occurs in postmitotic cells of the sensory thalamus and does not occur in the retina or in the cortex at early developmental stages (Chou et al., 2013). Three independent pools of mutant and control dissected visual thalami for each mutant line were processed. We identified 2202 differentially expressed genes (DEG) with FDR<0.01 following *Sox2* conditional inactivation, of which 981 downregulated and 1221 upregulated (Fig. 1C; Table S1). In addition, with the same thresholds, we identified 1010 genes differentially expressed following *Nr2f1* conditional inactivation, of which 474 genes downregulated and 536 upregulated (Fig. 1C; Table S1). Notably, *Sox2* shows only a slight decrease in transcript level in *Nr2f1* mutants, and *Nr2f1* shows an even slighter reduction in *Sox2* mutants (Fig. 1D; Table S1, in agreement with Mercurio et al., 2019a). This suggests that the leading cause of defects in *Sox2* and *Nr2f1* mutants is not the failure of SOX2 to activate *Nr2f1*, or the failure of NR2F1 to activate *Sox2*. This was also confirmed by immunofluorescence (IF) showing no changes in the number and distribution of NR2F1-expressing cells in the absence of SOX2 (Fig. S1B), suggesting that the two proteins do not importantly regulate each other.

### Many genes are co-regulated by SOX2 and NR2F1

We previously showed that important effectors of SOX2 function in neural stem cells (NSC) self-renewal and differentiation were among the most highly expressed and the most highly downregulated genes in *Sox2*-mutated cells (Bertolini et al., 2019; Pagin et al., 2021a,b). Importantly, genes dysregulated in *Sox2* mutant dLGN strikingly differed from those dysregulated in NSC (Bertolini et al., 2019). We thus focused on the most down- or upregulated genes in both thalamic mutants (Tables 1, 2: Tables S2, S3). We first asked whether there are common genes

deregulated in both *Sox2* and *Nr2f1* thalamic mutants. Fig. 1C shows that 514 genes significantly change their expression levels in both mutants, nearly all in the same direction (up, or down). This figure exceeds more than four times the number of differentially expressed genes expected by pure chance, i.e. if the *Sox2* and *Nr2f1* experiments were independent from one another (more than four times the expected value for a random overlap, and probability of having a similar overlap by chance<$10^{-100}$). Moreover, almost all the 50 genes in *Nr2f1* mutants with the most relevant decrease of transcript levels are also significantly downregulated in *Sox2* mutants (44 out of 50); similarly, most of the 50 genes in *Sox2* mutants with strongest decrease in transcript levels are significantly downregulated also in *Nr2f1* mutants (30 out of 50) (Tables 1, 2). Overall, these results clearly point to a SOX2 and NR2F1 co-regulated gene regulatory network in the early postnatal visual thalamus.

### Genes regulated by SOX2 and NR2F1 are enriched in functions related to neuronal differentiation and connectivity

To get insights into the collective functions of dysregulated genes, we performed Gene Ontology (GO) and functional enrichment analyses, focusing our attention on the genes significantly varying their expression in both *Sox2* and *Nr2f1* mutants (Fig. 1E). Results showed a striking enrichment in functions related to neuronal differentiation, and in particular to neuronal connectivity, activity and synaptic plasticity (Fig. 1E). Among the DEG annotated with these categories, regulators in axon guidance (Efna5, EphA5, EphA7, Sema7A), specifically in retinal axon guidance (Nrp1, Alcam, EphB1), in glutamatergic synapses (Grid1, Cdh8), as well as transcription factors involved in visual development and axonogenesis (Sox5), could be identified.

Notably, among genes significantly downregulated in both *Sox2* and *Nr2f1* mutants, *Vgf* deserved a particular attention, as it represented the most highly expressed, and one of the most strongly downregulated genes identified in both mutants (Tables 1, 2). *Vgf* encodes a diffusible cytokine, transported along thalamo-cortical axons until the axon terminals. Its function is important for the development of cortical layer 4, onto which it acts instructively to maintain the appropriate numbers of layer 4 neurons, in particular within the somatosensory and visual cortical areas (Sato et al., 2022).

*Sox5*, encoding a transcription factor key to the development of cortical neuronal connectivity (Kwan et al., 2008; Lai et al., 2008), was also strongly downregulated in both mutants (Tables 1, 2).

### CUT&RUN identifies SOX2 binding sites in the P0 visual thalamus

We set out to determine whether the functionally relevant deregulated genes are direct targets of SOX2 by establishing the genome-wide binding profile of this transcription factor. To overcome the anticipated technical difficulties due to the scant cell number obtained by dissection of this structure, we pooled dLGNs from P0 wild-type newborns and subjected them to CUT&RUN targeting SOX2 (Skene and Henikoff, 2017) (Fig. 2A). Several thousand SOX2 peaks were identified in two independent experiments; their overlap further defined a group of 717 high-confidence, reproducible binding events (Fig. 2B). SOX2 peaks were found in promoter and intronic, as well as intergenic regions, consistent with the genomic binding pattern identified in other cellular systems (Bertolini et al., 2019; Pagin et al., 2021b) (Fig. 2C). Two types of global analyses supported the dataset: a) motif analysis revealed SOX2 as the top enriched motif, along with other SOX factors, likely due to motif similarities (Fig. 2D),

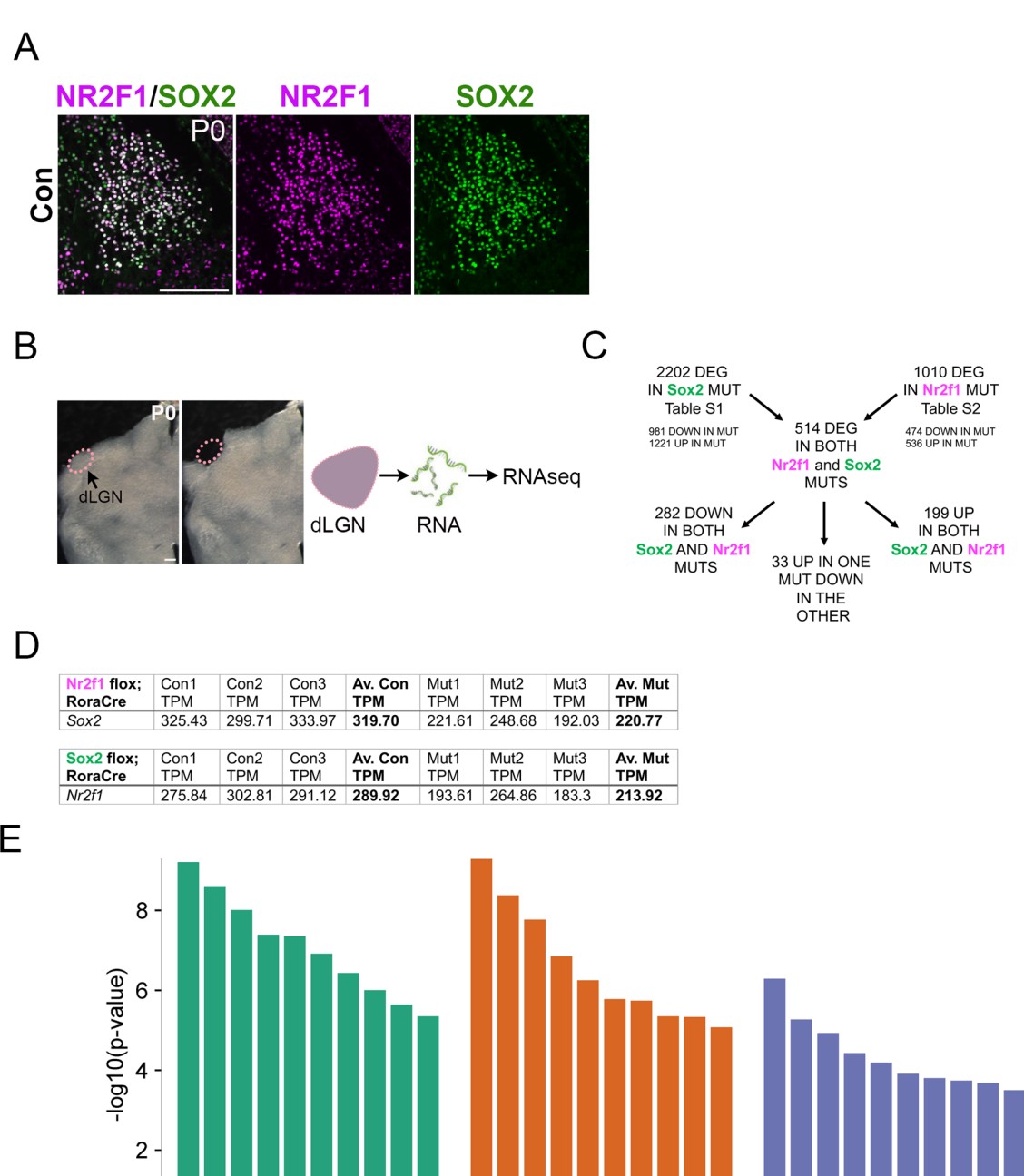

**Fig. 1.** See next page for legend.

**Fig. 1. RNA-seq of the visual thalamus (dLGN) of *Sox2* and *Nr2f1* thalamic mutants identifies many genes differentially expressed in both mutants.** (A) dLGN immunofluorescence with antibodies recognizing SOX2 (green) and NR2F1 (magenta) at P0 (wild type, con). Scale bar: 200 µm. (B) Sections comprising the dLGN at P0 used for RNA-seq analyses, before (left) and after (right) dissection. Scale bar: 200 µm. (C) Numbers of significantly dysregulated genes, identified in RNA-seq experiments, in *Sox2* and *Nr2f1* mutants, and in both mutants are shown. (D) Table showing *Sox2* RNA levels (TPM) in dLGN of *Nr2f1* mutants and controls (*Nr2f1^flox/flox^*, *Nr2f1^flox/+^*) and *Nr2f1* RNA levels (TPM) in dLGN of *Sox2* mutants and controls (*Sox2^1flox/flox^*, *Sox2^flox/+^*) (TPM for each biological replicate for controls and mutants are shown). (E) GO analysis of genes differentially expressed in both *Sox2* and *Nr2f1* mutants (DEG; the 514 genes in B) reveals enrichment in categories involved in neuronal development. The GO Biological Processes, Cellular Component and Molecular Function categories that are significantly enriched within the indicated mutants are shown.

increasing our confidence of the specificity of the identified peak regions. b) Gene ontology of peak-associated genes (by GREAT, McLean et al., 2010) revealed enrichment for primarily neuronal biological processes (Fig. 2E), supporting involvement in the development of neurons and their connectivity.

We compared the genome-wide physical occupancy of SOX2 in the dLGN to our previously identified binding of this factor in brain-derived NSC performed both with CUT&RUN and with ChIP-seq (Bertolini et al., 2019; Pagin et al., 2021b). Among the 717 high-confidence targets, 248 were shared with NSC (Fig. 2F). This allowed us to define a subset of 469 dLGN specific SOX2 targets (Fig. 2F; Fig. S3A), which likely underlay a different role of SOX2 in this differentiated cell population. Of note, while the shared targets almost only displayed motif enrichment for SOX factors (Fig. 2G), the dLGN-specific SOX2 peak regions contained an enrichment for the RORα consensus sequence, highly similar to the NR2F1 binding site – a motif that is not enriched in the NSC subset (Fig. 2H). Examples of peaks shared with the NSC and unique to the dLGN can be seen in Fig. 2I,L; note the presence of peaks in the proximity of the *Vgf* (Fig. 2L) and in the intron within *Sox5* (Fig. 2L) regions, which are among the most strongly downregulated genes in *Sox2* mutants (see above and Table 1), suggesting their direct regulation by SOX2.

GREAT associated the 717 binding sites with a total of 1102 genes (Fig. 2J). This allowed us to overlap our CUT&RUN with the genes identified to be expressed or differentially expressed when *Sox2* is inactivated (RNA-seq data, Fig. 1C; Table S1). We found that 784 of the 1102 peak-associated genes were expressed in dLGNs at P0 (TPM>5). Of these, 237 (145 down, 92 up) were dysregulated in SOX2 mutant dLGNs (<0.01 FDR) (Fig. 2J), providing robust evidence of SOX2 direct regulation in differentiated neurons.

To understand whether these genes could also be regulated by NR2F1, we overlapped these differentially expressed SOX2 targets with the genes dysregulated in *Nr2f1* mutant dLGNs. This led to a core signature of 79 SOX2 and NR2F1 coregulated genes, which display known literature-based physical and functional interactions (Fig. 2K, STRING Szklarczyk et al., 2021 diagram of core interacting SOX2/NR2F1 coregulated genes, disconnected nodes removed). Of note, this list includes several genes that are among the top 100 up- or downregulated upon SOX2 deletion in the dLGN, including *Sgk1*, *Sox5*, *Gsg1l*, *Frrs1l*, *Sox13*, *Neurog2* (Fig. 2K, Table S1).

## Deconvolution analysis identifies specific cell types affected by *Sox2* or *Nr2f1* deletion

A recent single-cell RNA-seq (scRNA-seq) analysis of the postnatal developing visual thalamus (Kalish et al., 2018), defined specific cell types by their distinct transcriptional identity. Specific gene

**Table 1. Genes most downregulated in *Sox2*-mutant visual thalamus**

| Gene | Av. Exp. | Av. Exp. | Av. Exp. Sox2 WT/ Av. Exp. Sox2 MUT | log fold change | DEG in Nr2f1 MUT |
|---|---|---|---|---|---|
| 9030625G05Rik | 8.1 | 1.05 | 7.69 | −2.90 | **YES** |
| Hdc | 21.82 | 2.97 | 7.35 | −2.81 | |
| Fam83f | 9.84 | 1.6 | 6.15 | −2.55 | YES |
| Ptpn22 | 4.52 | 0.84 | 5.4 | −2.37 | YES |
| Tuba8 | 9.12 | 1.75 | 5.2 | −2.31 | |
| Hs6st2 | 109.4 | 22.54 | 4.85 | −2.23 | |
| Pcdhac1* | 14.92 | 3.04 | 4.9 | −2.22 | **YES** |
| Drd5 | 10.05 | 2.23 | 4.51 | −2.11 | YES |
| Flt3 | 11.74 | 2.72 | 4.31 | −2.04 | YES |
| Wnt9b | 50.83 | 12.19 | 4.17 | −2.00 | **YES** |
| Hs3st1 | 121.89 | 29.48 | 4.13 | −1.99 | |
| Rora | 62.54 | 16.24 | 3.85 | −1.96 | |
| Nexn | 7.58 | 1.92 | 3.95 | −1.93 | **YES** |
| Neurog2 | 12.08 | 3.05 | 3.96 | −1.93 | **YES** |
| Hs6st3* | 48.32 | 12.18 | 3.97 | −1.92 | **YES** |
| Vgf | 882.88 | 231.32 | 3.82 | −1.87 | **YES** |
| Muc15 | 6.6 | 1.72 | 3.84 | −1.87 | **YES** |
| Sp9* | 269.34 | 78.82 | 3.42 | −1.71 | |
| Lgi2 | 35.57 | 10.41 | 3.42 | −1.71 | |
| Slc18a2 | 218.62 | 64.43 | 3.39 | −1.70 | |
| Col9a1 | 9.37 | 3 | 3.13 | −1.70 | |
| Cpne9 | 43.55 | 13.27 | 3.28 | −1.66 | YES |
| Efna5 | 37.94 | 11.68 | 3.25 | −1.64 | |
| Cd47 | 129.27 | 41.23 | 3.14 | −1.59 | YES |
| Tmem132d | 20.26 | 6.61 | 3.07 | −1.55 | **YES** |
| Galnt14 | 34.04 | 11.15 | 3.05 | −1.55 | |
| Drd1 | 4.73 | 1.55 | 3.04 | −1.54 | |
| 2900005J15Rik | 6.47 | 2.13 | 3.04 | −1.54 | |
| Chrm3 | 16.91 | 5.67 | 2.98 | −1.51 | **YES** |
| Adgrl2 | 88.59 | 29.81 | 2.97 | −1.51 | YES |
| Tmem132b* | 70.73 | 23.74 | 2.98 | −1.51 | **YES** |
| Edaradd | 9.05 | 3.05 | 2.97 | −1.51 | |
| Frrs1l | 71.66 | 24.07 | 2.98 | −1.51 | YES |
| Zmat4 | 25.97 | 8.78 | 2.96 | −1.50 | **YES** |
| Kcnn1 | 45.79 | 15.68 | 2.92 | −1.48 | |
| Tmem163 | 142.8 | 48.96 | 2.92 | −1.48 | YES |
| Ephx4 | 9.52 | 3.3 | 2.88 | −1.47 | YES |
| Trhde | 12.26 | 4.25 | 2.89 | −1.46 | |
| Sox13 | 86.24 | 30.48 | 2.83 | −1.44 | YES |
| Cdca7 | 51.7 | 18.3 | 2.83 | −1.44 | |
| Dmrtb1* | 111.05 | 39.94 | 2.78 | −1.42 | **YES** |
| Cpne4 | 34.3 | 12.39 | 2.77 | −1.41 | **YES** |
| Slc6a4 | 58.61 | 21.26 | 2.76 | −1.40 | **YES** |
| Fzd8 | 17.56 | 6.36 | 2.76 | −1.40 | |
| Sox5 | 25.07 | 9.1 | 2.75 | −1.40 | |
| Dkk4 | 6.33 | 2.33 | 2.71 | −1.37 | **YES** |
| Il22 | 7.01 | 2.6 | 2.69 | −1.37 | |
| Camk4 | 88.09 | 32.64 | 2.7 | −1.37 | YES |
| Gm15417 | 17.04 | 6.48 | 2.63 | −1.35 | YES |
| Dusp5 | 8.72 | 3.31 | 2.63 | −1.34 | |

expression signatures characterizing excitatory neurons, inhibitory neurons, oligodendrocytes, astrocytes, endothelial cells, pericytes and microglia could be defined, in a wild-type condition. We thus performed a deconvolution analysis of our RNA-seq data, based on these single-cell transcriptional profiles at P5, the earliest time point analyzed by the authors (Kalish et al., 2018) (Fig. 3), in order to estimate the abundance of each cell type in our samples. This showed a relevant reduction of the estimated fraction of glutamatergic neurons, in both *Sox2* and (to a lesser extent) *Nr2f1* mutant dLGNs (Fig. 3A,B; Table S4). Interneurons, pericytes and endothelial cells, conversely, were slightly increased, in both mutants. Oligodendrocytes were slightly reduced in *Sox2* mutants

**Table 2. Genes most downregulated in *Nr2f1*-mutant visual thalamus**

| Gene | Av. Exp. | Av. Exp. | Av. Exp. **Nr2f1** WT/Av. Exp. **Nr2f1 MUT** | log fold change | DEG in **Sox2 MUT** |
|---|---|---|---|---|---|
| Cck | 592.67 | 105.75 | 5.6 | −2.49 | YES |
| Nexn | 5 | 0.95 | 5.28 | −2.42 | YES |
| Tmem132d | 27.05 | 5.84 | 4.64 | −2.22 | YES |
| Slc6a4 | 41.84 | 11.42 | 3.66 | −1.88 | YES |
| Fam163a | 30.74 | 8.83 | 3.48 | −1.81 | YES |
| Cpne4 | 44.25 | 13.24 | 3.34 | −1.75 | YES |
| Dlk1 | 38.85 | 12.04 | 3.23 | −1.73 | YES |
| Rasd1 | 9.85 | 3.13 | 3.15 | −1.65 | YES |
| Stxbp6 | 35.08 | 11.5 | 3.05 | −1.62 | YES |
| Vgf | 855.03 | 282.91 | 3.02 | −1.61 | YES |
| Zmat4 | 24.53 | 8.21 | 2.99 | −1.59 | YES |
| Wnt9b | 57.22 | 19.28 | 2.97 | −1.58 | YES |
| Dkk4 | 7.87 | 2.74 | 2.88 | −1.52 | YES |
| 9030625G05Rik | 4.6 | 1.62 | 2.83 | −1.51 | YES |
| Pex5l | 8.52 | 3.07 | 2.77 | −1.50 | YES |
| Epha7 | 34.01 | 11.79 | 2.89 | −1.44 | YES |
| Asic2 | 85.02 | 32.42 | 2.62 | −1.43 | YES |
| Hr | 14.71 | 5.53 | 2.66 | −1.42 | YES |
| Ak5 | 48.21 | 19.44 | 2.48 | −1.32 | YES |
| Ccer2* | 5.99 | 2.51 | 2.39 | −1.27 | YES |
| Itga9 | 8.92 | 3.89 | 2.3 | −1.21 | YES |
| Hs6st3* | 64.49 | 28.25 | 2.28 | −1.21 | YES |
| Fam189a1* | 95.17 | 43.11 | 2.21 | −1.16 | YES |
| Smyd1 | 46.41 | 21.22 | 2.19 | −1.14 | YES |
| Epha4 | 85.04 | 39.26 | 2.17 | −1.13 | YES |
| Abcc8 | 13.12 | 6.14 | 2.14 | −1.11 | YES |
| Cdh8 | 101.92 | 48.02 | 2.12 | −1.10 | YES |
| Nefm | 353.74 | 167.17 | 2.12 | −1.09 | YES |
| Chrm3 | 22.76 | 10.83 | 2.1 | −1.09 | YES |
| Pcdhac1* | 17.48 | 8.31 | 2.1 | −1.09 | YES |
| Ism1 | 4.45 | 2.16 | 2.06 | −1.06 | |
| Muc15 | 7.49 | 3.64 | 2.06 | −1.05 | YES |
| Trpc3 | 27.24 | 13.31 | 2.05 | −1.05 | YES |
| Ube2ql1 | 235.63 | 115.61 | 2.04 | −1.04 | YES |
| Chrna7 | 9.75 | 4.81 | 2.02 | −1.03 | |
| Mag | 13.17 | 6.51 | 2.02 | −1.03 | YES |
| Cntnap5a | 13.99 | 6.96 | 2.01 | −1.02 | YES |
| Tmem132b* | 94.79 | 47.41 | 2 | −1.02 | YES |
| Samd5* | 23.94 | 12.02 | 1.99 | −1.01 | YES |
| Gal | 25.19 | 12.54 | 2.01 | −1.01 | |
| Hs3st5 | 16.51 | 8.29 | 1.99 | −1.01 | YES |
| Dcdc2a | 7 | 3.54 | 1.98 | −0.99 | YES |
| Mef2c | 31.56 | 16.19 | 1.95 | −0.98 | |
| Sv2b | 9.49 | 4.99 | 1.9 | −0.95 | |
| BC089491 | 9.83 | 5.16 | 1.9 | −0.94 | |
| Cdh7 | 12.1 | 6.39 | 1.9 | −0.94 | YES |
| Dmrtb1* | 113.78 | 60.29 | 1.89 | −0.93 | YES |
| Vangl1 | 34.34 | 18.29 | 1.88 | −0.93 | YES |
| Neurog2 | 13.95 | 7.52 | 1.86 | −0.91 | YES |
| St3gal1 | 20.68 | 11.17 | 1.85 | −0.90 | YES |

Av. Exp., average expression; TPM, transcripts per million; DEG, differentially expressed gene.

*: T-dark genes (genes about whose function very little is known; see https://pharos.nih.gov/, https://pharos.nih.gov/targets). YES in column five marks those genes that are also significantly dysregulated in the 'other' mutant, i.e. *Nr2f1* mutant for tables listing genes dysregulated in *Sox2* mutants, and vice versa; YES genes vary, in the 'other' mutant, in the same direction (downregulated in Tables 1, 2 and upregulated in Tables S2, S3). **YES** (in bold) means that the gene is among the 50 most down- or upregulated genes also in the 'other' mutant.

and increased in *Nr2f1* mutants. Of note, these changes were detected prior to overt abnormalities in the cell type composition of the mutant thalami, as previously shown by immunofluorescence and *in situ* hybridization only a week later, at P7 and/or P8 (Mercurio et al., 2019a).

### *In situ* hybridization and immunofluorescence validate and better describe the reduction in expression of key shared targets of SOX2 and NR2F1, *Vgf* and *Sox5*

We wished to validate the reduction of expression of some important genes previously identified by RNA-seq. We focused our analyses on *Vgf* and *Sox5* because 1) they are biologically relevant in the thalamo-cortical connectivity (see above and Discussion); 2) their regulatory regions are bound by SOX2 in our CUT&RUN experiments (Fig. 2L). *In situ* hybridization (ISH) clearly validates a reduction in expression of both *Vgf* and *Sox5* in the dLGN of both *Sox2* and *Nr2f1* mutants at E18.5, a stage in which an obvious thalamic phenotype is not yet observed (Mercurio et al., 2019a) (Fig. 4A). This contrasts with the essentially unchanged expression levels of *Nr2f1* in *Sox2* mutants and *Sox2* in *Nr2f1* mutants (Fig. 4A). The very similar distribution of *Sox2* and *Nr2f1* mRNAs by ISH in wild types and mutants also confirms the comparable size of the dLGN in mutants and controls at this stage, in agreement with our previous findings (Mercurio et al., 2019a). IF further confirms the reduction in expression of SOX5 in both *Sox2* and *Nr2f1* mutants (Fig. 4B,C). ISH and IF further allow to appreciate the differential distribution of the gene products between wild types and mutants; for *Sox5* in particular we note a residual expression in the *Sox2* mutant in a small area, while in the *Nr2f1* mutant in a characteristic stripe adjacent to the vLGN (Fig. 4A,B).

### DISCUSSION

The conditional deletion of *Sox2*, and of *Nr2f1*, from the developing visual thalamus leads to important, significantly overlapping defects in the thalamus itself, as well as in the thalamus-connected visual cortex (Chou et al., 2013; Mercurio et al., 2019a). We show that more than 1000 genes are dysregulated in each of the two mutants. Among them, an important subset of genes downregulated in both *Sox2* and *Nr2f1* mutants points to the existence of a shared network of genes essential for the proper development of the dLGN and its connections. The limited decrease in transcript levels of *Sox2* in *Nr2f1* mutants and of *Nr2f1* in *Sox2* mutants (Fig. 1D) suggests that the gene regulatory network jointly regulated by SOX2 and NR2F1, rather than the regulation of *Nr2f1* by SOX2, or of *Sox2* by NR2F1, is important in the shared phenotype shown by both mutants. In principle, the genes downregulated in the *Sox2*- and the *Nr2f1*-mutant dLGNs should explain three types of phenotypic defects observed in mutant mice (Chou et al., 2013; Mercurio et al., 2019a): 1) the reduction in the number of neurons (glutamatergic) in the mutant visual thalami (at postnatal day 7); 2) the alterations in terms of amount and distribution of thalamo-cortical and cortico-thalamic connectivity; 3) the alterations in the retino-geniculate connections.

Indeed, GO enrichment analysis of the genes dysregulated in both mutants reveals a highly significant enrichment of categories related to functions that develop abnormally in the mutants. These include, as the most significantly enriched categories, biological processes such as axonogenesis, neuron projection morphogenesis, chemical synapse transmission, axon development, retinal ganglion cell axon guidance, and related cellular components (Fig. 1E). It is likely that, collectively, genes with these functional annotations underlie, by their altered regulation, the above-mentioned defects.

### Mechanisms of dysregulation of gene expression in mutant thalamus

The observed dysregulation of several genes in the mutant thalami could be due to variations in the abundance of specific cell types, as well as to variations in gene expression levels within specific cell types normally expressing SOX2 and NR2F1. They may also be due

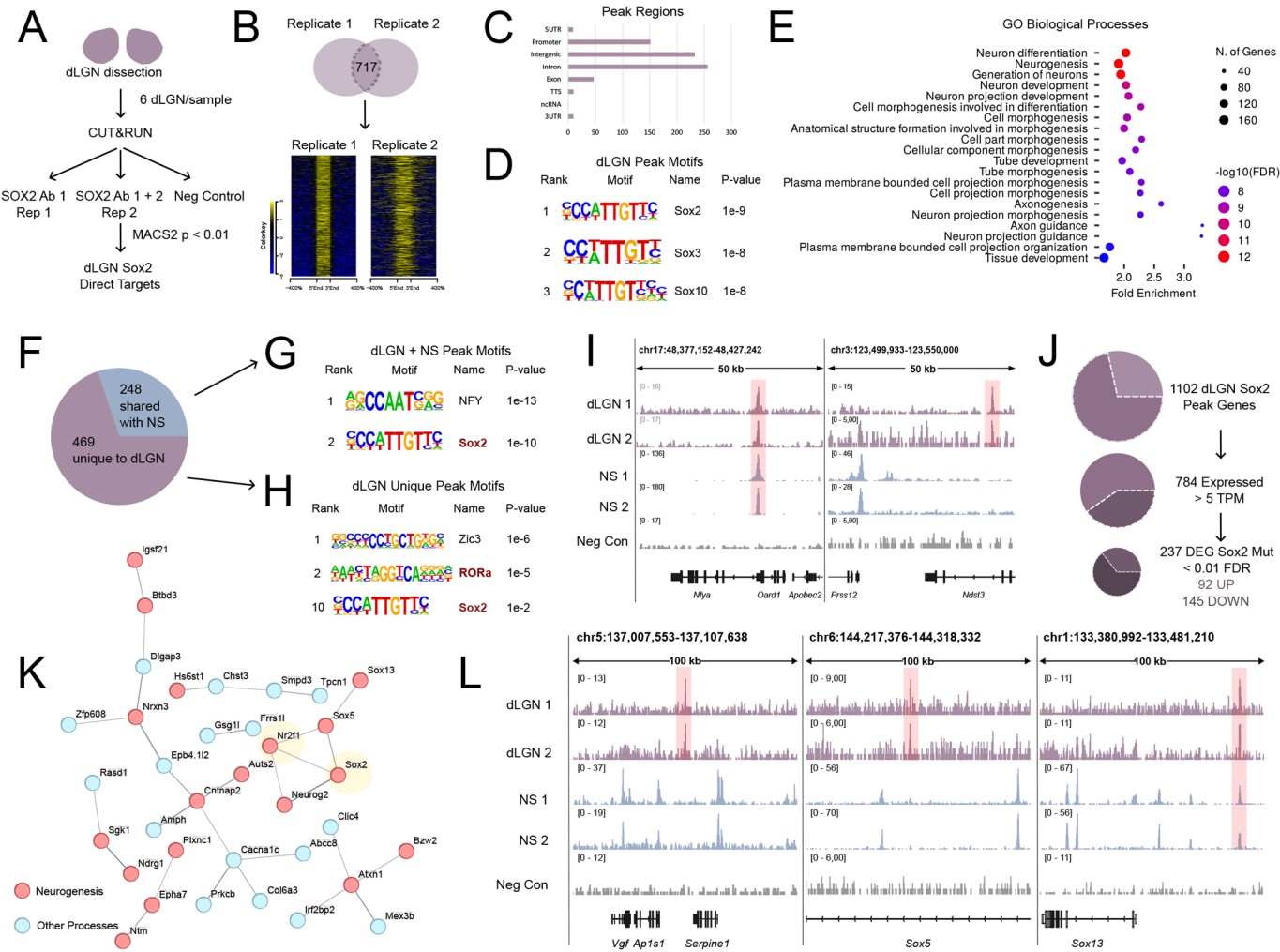

**Fig. 2. CUT&RUN of SOX2 binding in the P0 visual thalamus.** (A) Schematic depiction of the CUT&RUN experimental design. Two independent biological replicates for SOX2 and an anti-HA negative control were performed from pools of six dorsal lateral geniculate nuclei (dLGN) from three brains. (B) Venn diagram overlap and signal intensity plots of the two SOX2 replicates, showing enrichment over the control in all peak regions. (C) Peak region annotation by HOMER, showing that SOX2 binds primarily promoter, intronic and intergenic regions in dLGN. (D) HOMER known motifs for dLGN SOX2 peaks. The SOX2 motif is the most highly ranked, followed by other SOX factors. (E) GO enrichment of biological processes for genes associated by GREAT to SOX2 dLGN peaks. Dot size shows number of peak associated genes, dot color represents –log10 FDR (FDR<0.05), and the x axis represents fold enrichment. The top 20 terms are shown. Enrichment terms include neuron development related processes. (F) Pie chart depicting the distribution of SOX2 dLGN peaks in unique peaks and those shared with NSC SOX2 datasets. (G) HOMER known motifs for dLGN and NS shared peaks. NFY and SOX2 are the top motifs. (H) HOMER known motifs for dLGN unique peaks. Top motifs include ZIC3 and RORα. Sox2 is ranked 10th. (I) CUT&RUN tracks as visualized in Integrative Genome Viewer (IGV), showing both dLGN and NS shared peaks (left) and dLGN unique peaks (right). (J) Schematic depiction of CUT&RUN and RNA-seq overlap, showing Sox2 peak associated genes that are transcribed (>5 TPM, 784/1102) and those that are differentially expressed (DEG) in *Sox2* mutant dLGN (FDR<0.01, 327/784), and those that are up- (92) or downregulated (145) in the mutant. (K) STRING map of interactions between SOX2, NR2F1, and the 79 SOX2 dLGN direct targets that are dysregulated in both *Sox2* and *Nr2f1* mutant mice. Confidence was set on default (0.4), disconnected nodes were removed, and line thickness represents confidence. Red nodes are those known to be involved in neurogenesis, while blue nodes are not included in the neurogenesis set and could represent novel genes important to the generation of neurons in the visual thalamus. (L) CUT&RUN tracks showing SOX2 dLGN peaks near the important targets *Vgf* (left), *Sox5* (center, intronic region), and *Sox13* (right). While the peaks near *Vgf* and *Sox5* are unique to the dLGN datasets, the *Sox13* peak is also bound by SOX2 in NS.

to indirect effects, whereby decreased expression of genes downregulated following *Sox2* or *Nr2f1* knockout affects the expression of other genes not directly targeted by SOX2 or NR2F1 themselves. Evidence for direct effects of *Sox2* or *Nr2f1* deficiency is provided by CUT&RUN, which identified several hundreds of chromatin sites bound by SOX2 in replicate experiments (Fig. 2).

The variation in cell numbers in the mutants is indeed important. In fact, in previous work, we detected a reduction in the number of neurons by P7 following *Sox2* thalamic deletion (Mercurio et al., 2019a). In the present work, we detect a specific reduction in the fraction of cells bearing a transcriptional signature characteristic of

projection neurons, already at P0 (Fig. 3, deconvolution). This reduction is likely to be rooted in gene expression abnormalities within glutamatergic neurons themselves, which normally express SOX2 and NR2F1, and not in a decreased number of cells. Accordingly, several of our downregulated genes (e.g. *Vgf*, *Sox5*, *Rorα*) are reported to be prevalently expressed in neurons in published datasets of scRNA-seq analyses (Franzen et al., 2019). On the other hand, the large relative increase in GABAergic interneurons (that do not express SOX2 in the thalamus) (Mercurio et al., 2019a), might be due to indirect effects, such as a change in the fate of part of the mutant neurons, or an increased interneuron

A

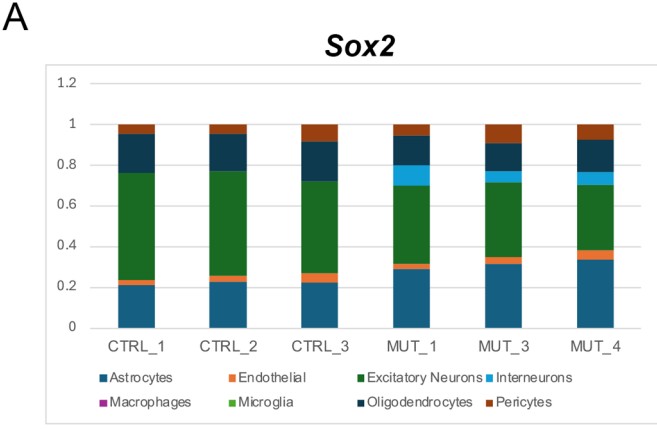

B

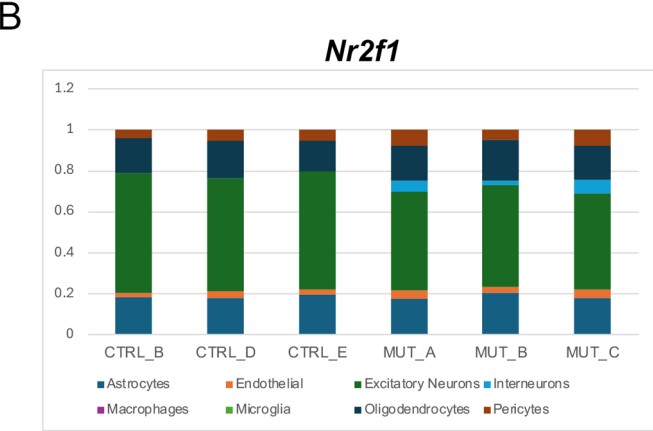

**Fig. 3. Deconvolution analysis documents the loss of projection neurons transcriptional identities in *Sox2* and *Nr2f1* mutants.** Cell type abundance variation in *Sox2* mutant and control (*Sox2flox/flox* or *Sox2flox/+*) samples (A) and *Nr2f1* mutant and control (*Nr2f1flox/flox* or *Nr2f1flox/+*) samples (B) expressed as fraction of cells of a given type gained or lost with respect to the average abundance across all samples considered. The cell types considered are those defined by their scRNA-seq transcriptional identity in (Kalish et al., 2018). Data (estimated fraction of each cell type) for all three mutant and control samples are shown numerically in Table S4 and graphically in A, B.

colonization of the mutant thalamus, or increased survival of the interneurons in the mutant thalamus.

### dLGN-specific SOX2 binding sites differ from those shared with neural stem cells

The comparison of SOX2 binding sites detected by CUT&RUN in the visual thalamus (present study) with those previously observed in neural stem cells by both CUT&RUN and ChIPseq (Bertolini et al., 2019; Pagin et al., 2021b) allowed us to compare the most represented SOX2 binding sequence motifs specific to the thalamus with those specific to NSC. SOX2 binding sites are among the most represented sites in both datasets, as expected (Fig. 2D,G,H). Remarkably, however, the thalamus-specific SOX2-binding sites showed, among the most represented, the binding site for RORα, also called NR1F1. As discussed above, NR1F1 and NR2F1, highly homologous, recognize similar DNA sequences with a core reported to be identical (https://www.genecards.org/cgi-bin/carddisp.pl?gene=RORA; https://www.genecards.org/cgi-bin/carddisp.pl?gene=NR2F1). This suggests that the molecular basis for the sharing of co-regulated target genes between SOX2 and NR2F1

might lie in the co-binding of the two factors to regulatory DNA sequences. Of note we detect direct SOX2 binding sites within putative regulatory regions of some of our most downregulated and biologically significant SOX2 targets (*Sox5* and *Vgf*, Fig. 2L).

### Potential key contributors of SOX2- and NR2F1-dependent vision development

As shown in Tables 1, 2, a large proportion of genes highly expressed in wild-type cells, and highly downregulated in *Sox2* mutants, are also downregulated in *Nr2f1* mutants; the reciprocal is also true. In contrast, among the genes upregulated in *Sox2* mutants, just a few are upregulated also in *Nr2f1* mutants (Tables S2, S3). This suggests that the functional association of SOX2 and NR2F1 is mostly involved in positive regulation (activation) of gene transcription.

### Vgf

Among the genes at the top of the list, *Vgf* is the most highly expressed and is among the ones with the largest decrease in transcript levels in both *Sox2* and *Nr2f1* mutants (Table 1). ISH fully validates this decrease in both mutants (Fig. 4A). *Vgf* encodes a signaling molecule, transported along thalamo-cortical axons until the axon terminals in the layer 4 of the cortex, where it acts instructively to maintain the appropriate neuron numbers within the visual and somatosensory cortical areas (Sato et al., 2022). The role of VGF in the layer was initially demonstrated by ablating the thalamo-cortical neuronal connections, thus depriving the cortex of sufficient VGF. Further, the knock-out of *Vgf* reduced layer 4 in both the somatosensory and visual cortices. This phenotype could then be rescued to normality by transgenic expression of VGF in the developing cortex (Sato et al., 2022). We observe (Fig. S2) that *RORβ*, marking cortical layer 4, is moderately reduced in the visual cortex of *Sox2* mutants, mirroring, at least to an extent, what is found in *Vgf* mutants (Sato et al., 2022). Knowing that VGF expression is strongly reduced in the mutant dLGN (Fig. 4A; Tables 1, 2), and that axonal connections reaching the visual cortex are also reduced in both mutants (Chou et al., 2013; Mercurio et al., 2019a), and in agreement with the abnormalities in the primary (V1) and higher order (V^HO) visual cortical areas in both thalamic mutants (Chou et al., 2013; Mercurio et al., 2019a), we propose that a reduced amount of VGF reaching the axon terminals in the visual cortex may contribute to the observed cortical layer defects of the *Sox2* mutant (Mercurio et al., 2019a and Fig. S2). Thus, VGF might be a mediator of the function of SOX2 during area- and layer-specific cortical development. Furthermore, *Sox2* is also downregulated in the somatosensory VP (Ventro-Posterior) thalamic nucleus in *Sox2* thalamo-specific conditional mutants, leading to moderate abnormalities of the somatosensory cortical area (Mercurio et al., 2019a). We thus looked at the somatosensory cortex layer 4 and observed a *RORβ* reduction (less marked than in the visual area) also in this area (Fig. S2), in agreement with the above hypothesis. We note, however, that the reduction of thalamo-cortical afferents in general rather than just the contribution of VGF maybe contribute to the observed phenotypes.

### Sox5

SOX5, a transcription factor, is significantly downregulated in both mutants at the mRNA and protein level (Tables 1, 2; Fig. 4A-C). ISH and IF fully validate a reduction of expression in both mutant dLGNs and highlight that the residual SOX5 positive cells in the mutants show a peculiar distribution different from that of the wild type (Fig. 4A,B). In humans, the *SOX5* gene

Biology Open

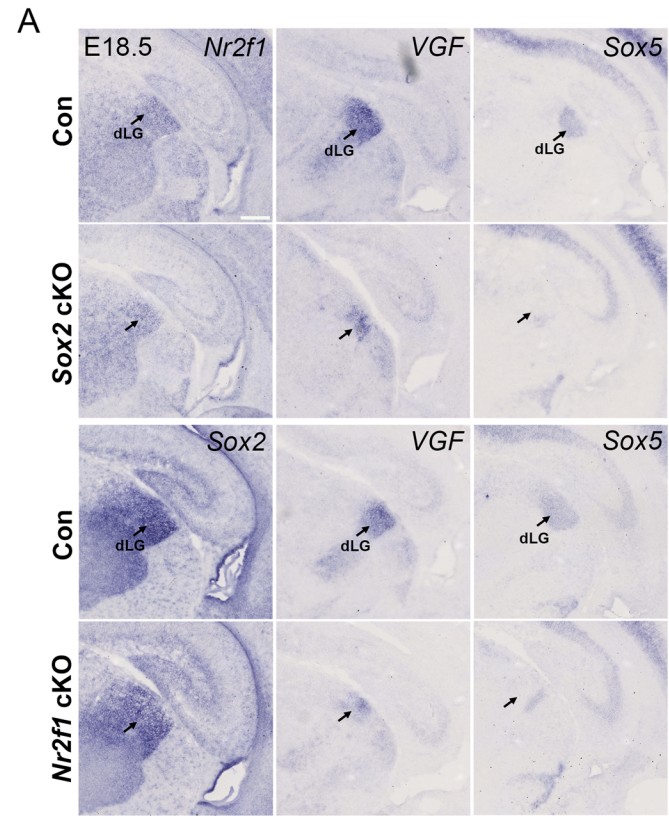

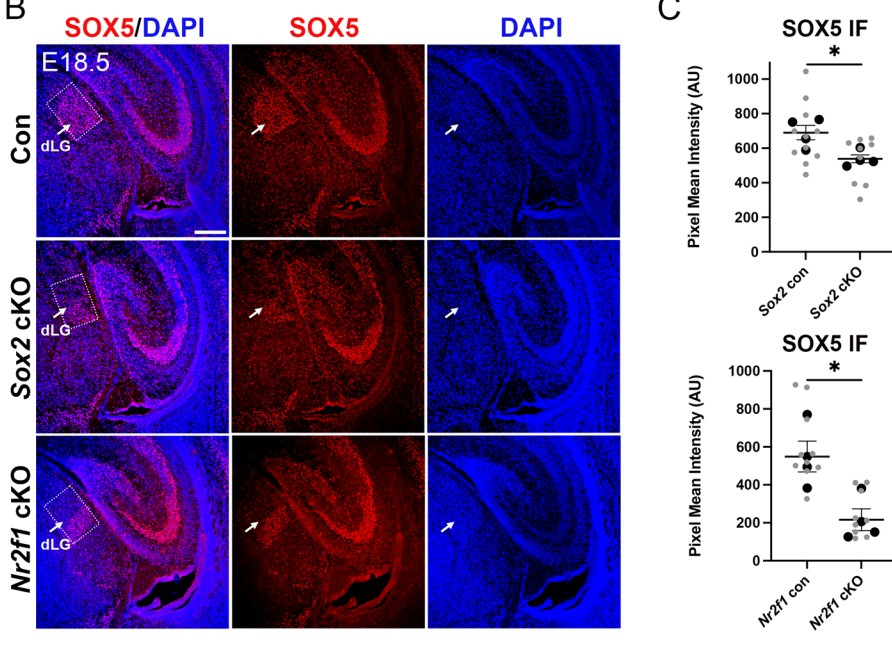

**Fig. 4. ISH and immunofluorescence document *Vgf* and *Sox5* downregulation in *Sox2* and *Nr2f1* thalamic mutants.** (A) Top, ISH with *Nr2f1*, *Vgf* and *Sox5* probes on coronal sections of E18.5 brains of *Sox2* mutants (*Sox2* cKO) and control littermates (*Sox2^{flox/flox}* or *Sox2^{flox/+}*). Bottom, ISH with *Sox2*, *Vgf* and *Sox5* on coronal sections of E18.5 brains of *Nr2f1* mutants (*Nr2f1* cKO) and control littermates (*Nr2f1^{flox/flox}* or *Nr2f1^{flox/+}*). At least four mutants and five controls were analyzed with each probe. Arrows indicate the dLGN. Note reduced expression of *Vgf* and *Sox5* in the dLGN of both mutants compared to the respective controls. *Nr2f1* and *Sox2* appear very slightly affected in the *Sox2* mutant and *Nr2f1* mutant, respectively. Scale bar: 200 μm. (B) IF with antibodies recognizing SOX5 (red) on coronal sections of E18.5 brains of controls (*Sox2^{flox/flox}*, *Sox2^{flox/+}*; *Nr2f1^{flox/flox}*, *Nr2f1^{flox/+}*), *Sox2* CKO and *Nr2f1* CKO. DAPI (blue) marks nuclei. At this stage, the dLGN is only modestly reduced in the mutants (Chou et al., 2013; Armentano et al., 2007). Arrows point to SOX5-positive cells within the dLGN region. Note the reduction of the SOX5-positive area in the *Sox2* and *Nr2f1* mutant dLGN. SOX5 positivity is not altered in the adjacent hippocampal region in mutants. Scale bar: 200 μm. (C) Graphs show the quantification of SOX5 pixel intensity within the dashed rectangles in B, in controls, *Sox2* cKO and *Nr2f1* cKO dLGNs. Large dots represent the mean intensity for each brain analyzed and the small dots represent each individual measurement. Mean±s.e.m. is indicated. *$P<0.05$, **$P<0.01$, ***$P<0.001$; unpaired *t*-test. The results shown are representative of $n=4$ mutants and $n=4$ control brains analyzed.

is associated with "Optic nerve hypoplasia bilateral, autosomal dominant" (OMIM #165550). As shown in *Sox5*-null mice (Lai et al., 2008; Kwan et al., 2008) this gene controls important aspects of neuronal connectivity, such as the development of cortico-fugal neurons, including cortical neurons projecting to thalamic neurons. Misrouting of subplate and layer 6 cortico-thalamic axons to the hypothalamus is also observed (Kwan et al., 2008). These data suggest the possibility that the reduction of SOX5 in the dLGN of *Sox2* and *Nr2f1* mutants affects axonogenesis.

## Hs6st2, Hs6st3

The Heparan sulphate 6-O-sulfotransferase enzyme modifies the sulfation status of heparan sulphate proteoglycans (HSPG) (Xu and Esko, 2014), extracellular matrix proteins, with covalently linked polysaccharidic chains, which are polymerized by specific enzymes and further modified by sulfation to obtain ample structural and functional diversity (Kreuger and Kjellen, 2012). HSPG interact with a variety of proteins, implicated in cell proliferation and differentiation, adhesion, migration, and other processes (Xu and Esko, 2014). It is intriguing that three of the genes (*Hs6st1, 2* and *3*)

encoding isoforms of the enzyme are downregulated in *Sox2* and/or *Nr2f1* mutant dLGNs. Axon guidance or axon extension defects abnormalities, and eye defects, have been reported in association with mutation of *Hs6st* genes (Xu and Esko, 2014; Kreuger and Kjellen, 2012; Habuchi and Kimata, 2010; Pratt et al., 2006; Tillo et al., 2016). We hypothesize that SOX2- and NR2F1-dependent expression of HS6ST enzymes, acting on the sulfation status of HSPGs, might play a role in the development of axonal connectivity of thalamic neurons.

### Rorα

*Rorα*, also named *Nr1f1*, is one of the genes with the largest decrease in transcript levels in *Sox2* mutants, although not in *Nr2f1* mutants (Tables 1, 2); yet, it may have a special significance also in the perspective of the present study, focusing on a SOX2 and NR2F1 co-regulated gene expression network. *Rorα* encodes a close homolog of NR2F1, and its product NR1F1 binds to a DNA sequence closely related to that of NR2F1, with an identical core binding site (https://www.genecards.org/cgi-bin/carddisp.pl?gene= RORA; https://www.genecards.org/cgi-bin/carddisp.pl?gene= NR2F1). *Rorα* spontaneous (staggered mouse) or engineered mutations cause ataxia and cerebellar neurodegeneration, with synaptic arrangement and immature morphology of cerebellar neurons (Purkinje) (Hamilton et al., 1996; Steinmayr et al., 1998). *RORα* defects are connected in humans with intellectual development disorder with or without epilepsy, or cerebellar ataxia, (OMIM #600825). If the activity of RORα on some genes partially overlaps with that of NR2F1, it is possible that, in the *Sox2* mutant, the reduced activity of RORα leads to downregulation of a set of genes closely related to those activated by NR2F1, explaining some common phenotypic defects observed in *Sox2* and *Nr2f1* mutants.

Overall, our work identifies a common transcriptional program driven by SOX2 and NR2F1 in the visual thalamus, highlighting a small subset of genes commonly downregulated in both mutants, and potentially important for explaining abnormalities of retina-thalamus-cortex connections. This paves the way to the precise identification, by functional transgenic studies, of genes mediating the common SOX2 and NR2F1 functions in the visual thalamus. It will also open the way to defining the molecular mechanisms mediating SOX2 and NR2F1 interactions in transcriptional regulation.

## MATERIALS AND METHODS

### Ethics statements

All procedures were in accordance with the European Communities Council Directive (2010/63/EU and 86/609/EEC) regulating animal research, and the Italian Law for Care and Use of Experimental Animals (DL26/14). They were approved by the Italian Ministry of Health (authorization no. 189/ 2016-PR to Prot. 29C09.4).

### Mouse strains

*Sox2* mutant mice were obtained by crossing *Sox2^flox* (Favaro et al., 2009) with *Rorα*-Cre (Chou et al., 2013) mouse lines. *Nr2f1* mutant mice were generated by crossing *Nr2f1^flox* (Armentano et al., 2007) with *Rorα*-Cre (Chou et al., 2013) mouse lines (note: COUP-TF1 is the old name of Nr2f1).

Genotyping was performed with the following primers (Chou et al., 2013; Mercurio et al., 2019a):

*Rorα*-Cre IRES Forward: 5′AGGAATGCAAGGTCTGTTGAAT 3′; *Rorα*-Cre IRES Reverse: 5′ TTTTTCAAAGGAAAACCACGTC 3′; *Sox2* flox Forward: 5′AAGGTACTGGGAAGGGACATTT 3′; *Sox2* flox Reverse: 5′AGGCTGAGTCGGGTCAATTA 3′; *COUP-TF1* flox Forward 5′-CTGCTGTAGGAATCCTGTCTC-3′; *COUP-TF1* flox Reverse:

5′-AATCCTCCTCGGTGAGAGTGG-3′ and 5′– AAGCAATTTGGCTT CCCCTGG-3′.

The day of vaginal plug was defined as embryonic day 0 (E0) and the day of birth as P0.

We note that *Rorα* is the gene to the 3′ of which the *Cre* gene has been inserted (as IRES-Cre knocked-in into the 3′UTR of the gene), in the Cre transgene driving *Sox2* and *Nr2f1* deletion in our thalamic mutants. In principle, *a priori*, the Cre insertion into the *Rorα* locus per se might have lowered *Nr1f1-Rora* expression. However, we also note that the significant (about four times) reduction of expression is observed in *Sox2* mutants, but not *Nr2f1* mutants, where expression levels are unchanged with respect to control. We thus think it is unlikely that Cre insertion plays a role in the expression reduction of *Rora* in Sox2 mutants. The *Rorα*-Cre and the *Sox2^flox* mouse lines are available at the European Mouse Mutant Archive (EMMA), EM:13253 and EM:07995, respectively. The *Nr2f1^flox* can be obtained upon request.

### Brain extraction and dissection for RNA-seq

Brains from *Sox2* thalamic mutants and controls (*Sox2^flox/flox* or *Sox2^flox/+*), and *Nr2f1* thalamic mutants and controls (*Nr2f1^flox/flox* or *Nr2f1^flox/+*), at P0 were removed from the skulls in ice-cold PBS1X, coronally embedded in low melt agarose 4% in PBS1X and kept on ice. Brains were then sectioned in ice-cold and sterile PBS1X with a vibratome (Leica VT1000s). 200 μm brain sections were collected in ice cold PBS1X. Sections including the dLGN were identified under a stereoscopic microscope and the dLGN was quickly dissected with sterile chirurgic scalpels on a glass slide. dLGNs were present usually in two sequential sections. Excised dLGNs were collected with a pipette, placed in sterile tubes, snap-frozen in liquid nitrogen and stored at −80°C until RNA extraction. All the sections were imaged before and after dissection of dLGNs.

### RNA extraction and RNA sequencing

RNA from snap-frozen dLGNs was extracted with the RNeasy Micro Kit (QIAGEN) with some precautions. Each dissected dLGN weighed about 0.3 mg. dLGNs samples were taken out of the −80°C and left a few seconds on the bench before tissue homogenization. Each dLGN sample was homogenized in 350 μl of RLT buffer containing β-mercaptoethanol (β-ME), as indicated by the manufacturer (10 μl of β-ME per 1 ml of RLT buffer). Tissue was homogenized by sequential passages through needles of descending diameter. Precisely we used a 1cc syringe and three different sterile needles: 18G, 22G and 26 G. First, the tissue was passed 10 times through the larger diameter needle (18 G), then 10 times through the intermediate diameter needle (22 G) and finally 10-12 times through the thinnest one (26 G), in order to correctly lysate and disrupt the tissue. Subsequently, the homogenized sample was vortexed for 30 s and centrifuged for 3 min at 13,200 rpm to pellet the debris. RNA was extracted from the homogenized supernatant as indicated by the manufacturer's instructions.

RNA sequencing was performed on three independent samples for both mutant and control dLGN. Each sample was composed of dLGNs from three animals of the same genotype pooled together. Genotypes were: for mutants, *Sox2^flox/flox*, or *Nr2f1^flox/flox*, plus *RORalpha*-Cre transgene; for controls, we used littermates of the respective mutants (*Sox2* or *Nr2f1* mutants), carrying two intact copies of *Sox2* (*Sox2^flox/flox* or *Sox2^flox/+*), or *Nr2f1* (*Nr2f1^flox/flox* or *Nr2f1^flox/+*), and no Cre transgene. For each sample sequenced we obtained at least 150 ng of high-quality total RNA (RIN≥8), and thus 150 ng were used for library preparation. Library preparation was performed with Nugen Universal+mRNA-seq kit, followed by sequencing in a HiSeq 4000 (Illumina), paired-reads 2x 75 bp. The KEGG 2019 Mouse database was used to analyze RNA-seq data.

The GO analysis reported in Fig. 1E was performed using Enrichr (https:// maayanlab.cloud/Enrichr).

### RNA-seq raw data analysis and deconvolution analysis

Sequence reads were mapped with STAR (Dobin et al., 2013) against the mouse RefSeq transcriptome, version April 2019, retrieved from the UCSC Genome Browser Database (Navarro Gonzalez et al., 2021). Read counts

and subsequent normalized transcripts per million (TPM) were computed with RSEM (rsem-calculate-expression) (Li and Dewey, 2011).

Differential expression analysis was performed with edgeR (Robinson et al., 2010). Initial read counts were normalized by trimmed mean of M values (TMM), with default parameters. Differentially expressed genes were identified by the quasi-likelihood (QL) *F*-test of edgeR (glmQLFfit and glmQLFtest functions, with default parameters). We selected as differentially expressed all genes with an adjusted *P*-value (FDR)<0.01.

Bulk RNA-seq deconvolution analysis was performed with MuSiC (Wang et al., 2019), with as input raw count tables for bulk RNA-seq and raw count tables at P5 for single cell RNA-seq samples. Single cell clusters and cell type annotations were retrieved from (Kalish et al., 2018).

For Tables 1 and 2, Tables S2 and S3, DEG were initially selected having expression levels higher than 4 TPM (at least in wild type, for downregulated genes; at least in mutant, for upregulated genes). Average expression values were then calculated from triplicate samples in Table S1. The top 50 most down- or upregulated genes (with greatest log-fold change in mutant versus wild type) are shown for both mutants (Tables 1, 2, Tables S2 and S3).

### Immunofluorescence

To detect SOX2 and NR2F1, slides were washed 2X for 10 min in PBS1X and then unmasked in Na Citrate 0.1M-Citric acid 0.1 M pH6 solution for 2 min. Sections were washed in PBS1X for 10 min and incubated 1 h with pre-blocking solution (sheep serum 5%, Triton 0,3% in PBS1X) at room temperature. Sections were then incubated over night at 4°C in blocking solution (sheep serum 1%, Triton 0,1% in PBS1X) containing primary antibodies. The following primary antibodies were used: anti-SOX2 diluted 1:500 (R&D AB2018, mouse) and anti-NR2F1 diluted 1:1000 (Abcam Ab181137, rabbit). Sections were then washed 2X for 10 min in PBS1X and incubated for 1 h at room temperature with blocking solution containing the following fluorescent secondary antibodies: anti-mouse IgG Alexa Fluor 488 and anti-rabbit IgG Alexa Fluor 594 (Thermo Fisher Scientific) diluted 1:500. Slides were then washed 2X for 10 min in PBS1X and mounted.

To detect SOX5, slides were treated as above, but without the unmasking step. The anti-SOX5 (Li et al., 2022) antibody [a gift from A. Morales, Instituto Cajal (CSIC), Madrid Spain] was diluted 1:500, and the anti-rabbit Alexa Fluor 594 (Thermo Fisher Scientific) was diluted 1:1000.

To measure the colocalization of SOX2 and NR2F1 in cells of the dLGN, the plane with the greatest signal intensity was chosen from the z-stack. A cortical reference region with strong NR2F1 signal was chosen as a control. Channels were split and read into the EzColocalization plugin (Stauffer et al., 2018), resulting in Pearson coefficient (PCC) for the correlation of SOX2 and NR2F1 in the dLGN versus the reference regions Fig. S1A. The PCC of two sections from the same animal were averaged. The values were tested positive for normal distribution with the Shapiro-Wilk test and positive for equal variances with the Levene's test. No outliers were identified using the grubb-test. Therefore, a paired two-sample *t*-test was performed (*n*=4). Superplot were generated with GraphPad, large dots represent the mean intensity for each brain analyzed and the small dots represent each individual measurement. Mean±s.e.m. is indicated. *P<0.05, **P<0.01, ***P<0.001.

To measure SOX5 IF signal in the dLGN of controls (*Sox2flox/flox*, *Sox2flox/+*, *Nr2f1flox/flox* or *Nr2f1flox/+*), *Sox2* cKO and *Nr2f1* cKO, images were acquired with a confocal microscope, pixel mean intensity was measured using ImageJ within a region of interest (ROI) placed on the dLGN, as shown in Fig. 4C. Background was measured in three different points in the sections, averaged and subtracted from the mean intensity. Two or three sections per brain were analyzed and their mean intensity averaged. Four control and mutant brains were analyzed for each cKO. An unpaired two-tailed *t*-test was performed. Superplot were generated with GraphPad, large dots represent the mean intensity for each brain analyzed and the small dots represent each individual measurement. Mean±s.e.m. is indicated. *P<0.05, **P<0.01, ***P<0.001.

### ISH

ISH was performed as previously described (Mercurio et al., 2019a). Briefly, brains at E18.5 were dissected and fixed overnight in paraformaldehyde (PFA) 4% in PBS (phosphate buffered saline) 1X. The

fixed tissue was cryoprotected in a series of sucrose solutions (15%, 30%) in PBS 1X and then embedded in OCT (Killik, Bio-Optica) and stored at −80°C. Brains were sectioned (20 μm) with a cryostat, placed on a slide (Super Frost Plus 09-OPLUS, Menzel) and stored at −80°C. Slides were then defrosted, fixed in formaldehyde 4% in PBS for 10 min (min), washed three times for 5 min in PBS 1X, incubated for 10 min in acetylation solution (for 200 ml: 2.66 ml triethanolamine, 0.32 ml HCl 37%, 0.5 ml acetic anhydride 98%) with constant stirring and then washed three times for 5 min in PBS1X. Slides were placed in a humid chamber and covered with prehybridization solution (50% formamide, 5X SSC, 0.25 mg/ml tRNA, 5X Denhardt's, 0.5 μg/ml salmon sperm) for at least 2 h and then incubated in hybridization solution [fresh prehybridization solution containing the digoxygenin (DIG)-labelled RNA probe of interest] overnight at 65°C. Slides were washed 5 min in 5X SSC, incubated twice in 0.2X SSC for 30 min at 65°C, washed 5 min in 0.2X SSC at room temperature and then 5 min in maleic acid buffer (MAB, 100 mM maleic acid, 150 mM NaCl pH 7.5). The slides were incubated in blocking solution [10% sheep serum, 2% blocking reagent (Roche), 0.3% Tween-20 in MAB] for at least 2 h at room temperature, then covered with fresh blocking solution containing anti-DIG antibody Roche 1:2000 and finally placed overnight at 4°C. Slides were washed in MAB three times for 5 min, in NTMT solution (100 mM NaCl, 100 mM Tris-HCl pH 9.5, 50 mM MgCl₂, 0.1% Tween-20) twice for 10 min and then placed in a humid chamber, covered with BM Purple (Roche), incubated at 37°C until desired staining was obtained (1-6 h), washed in water for 5 min, air dried and mounted with Eukitt (Sigma-Aldrich).

The DIG-labelled *RORβ* and *Sox5* probes were prepared as in (Nakagawa and O'Leary, 2003; Lioubinski et al., 2003). The *Vgf* DIG-labelled probes were generated from PCR generated templates that included the T7 promoter and were amplified from mouse P0 dLGN cDNA with the following primers:

*Vgf* F 5′ CGTCCTCTTCTGCTTCCTTCTA 3′
*Vgf* R+T7 5′ TAATACGACTCACTATAGGTTTTAGGGGAGGACA-CTCCTT 3′.

### CUT&RUN

CUT&RUN was performed as described previously (Skene et al., 2018). dLGNs were dissected (as described above in Brain extraction and dissection for RNA-seq), suspended in 1.5 ml of cold CUT&RUN wash buffer [HEPES pH 7.5 (20 mM), NaCl (150 mM), Spermidine (0.5 mM), Roche Complete Protease Inhibitor EDTA-Free (cat. #COEDTAFRO, Roche)] and then manually dissociated by gentle pipetting. Cells were collected by centrifugation at 600 *g* for 3 min, the supernatant carefully removed and then washed two more times by resuspension in 1 ml wash buffer. 4 μl of ConA bead slurry (prepared in binding buffer as described), were added per pair of dLGNs and the cells and beads were incubated for 10 min on a rotator. While the rest of the dissections proceeded, bead bound cells were kept in storage buffer [wash buffer with EDTA (2 mM)]. Three brains (six dLGNs) were collected per sample. Once all dLGNs were collected, the beads were collected on the magnet and resuspended in 150 μl antibody buffer [wash buffer with EDTA (2 mM) and 0.025% digitonin]. Antibodies were added at 1:100 dilution and incubation proceeded ON at 4°C. Antibodies used included anti-SOX2 (ABIN2855074, antibodies online), anti-SOX2 (ABIN2855073, antibodies online), and anti-HA (05-902R, Merck). One SOX2 sample was with only ABIN2855074, the other was with both SOX2 antibodies together. The two SOX2 biological replicates were performed independently from different litters of mice. The next day samples were washed twice with dig-wash (wash buffer with 0.025% digitonin) and then resuspended in 150 μl dig-wash containing pA-MNase (New England Biolabs 700 ng/ml, received as a gift from Steven Henikoff, Howard Hughes Medical Institute, Basic Sciences Division, Fred Hutchinson Cancer Research Center, USA) at 1:200 dilution and rotated for 1 h. Samples were washed twice and then resuspended in 100 μl dig-wash and equilibrated in ice. 2 μl 100 mM CaCl₂ was added, and digestion proceeded for 30 min in ice. 100 μl 2X STOP buffer [NaCl (340 mM), EDTA (20 mM), EGTA (4 mM), digitonin (0.05%), RNase A (100 μg/ml), glycogen (50 μl/ml)] was added to stop the digestion reaction and incubated at 37°C for 30 min. Samples were pelleted at 16,000 *g* for 5 min and placed on the magnet rack. The supernatant was harvested, and beads were discarded. DNA purification was performed with phenol chloroform. Library

preparation was performed using the KAPA HyperPrep Kit (cat. #KK8504, KAPA Biosystems) according to the manufacturer's instructions, using KAPA DUI adapters at 0.15 μM. Libraries were sequenced with the Illumina NextSeq 550 using the High-Output 75 cycles kit v2.5 (cat. #20024906, Illumina), 36 base pair pair-end.

## CUT&RUN data analysis

Reads were trimmed to remove adapters, artifacts and repeats of poly $[AT]_{36}$, $[C]_{36}$ and $[G]_{36}$ with bbmap bbduk (Bushnell et al., 2017) (version 38.18). Alignment was performed to the mm10 mouse genome with bowtie (Langmead et al., 2009) (version 1.0.0) with the options -v 1 -m 1 -X 500. Samtools (Li et al., 2009) (version 1.11) was used for deduplication and to remove and incorrectly paired reads. Bedtools (Quinlan and Hall, 2010) (version 2.30.0) was used to remove reads mapped to the CUT&RUN mm10 (Nordin et al., 2023) from bam files. Peaks were called for each replicate using MACS2 (Zhang et al., 2008) with the options -f BAMPE –keep-dup all -p 1e-2 -SPMR -bdg against the anti-HA negative control. Output narrowPeak files were overlapped using Bedtools intersect, keeping only reproducible peak regions called in both biological replicates. Signal intensity plots were created using ngsplot (https://doi.org/10.1186/1471-2164-15-284, version 2.63) with options -N 4 -GO none -SC global, plotting signal intensity of the replicates compared to the anti-HA control. Motif analysis was done using HOMER (Li et al., 2009) (version 4.11) findMotifsGenome to find motifs in the mm10 genome using -size given, and peak region annotation was done with HOMER annotatePeaks on default settings. Peak set gene annotation was done using GREAT (McLean et al., 2010) (version 4.0.4) with default parameters, and gene names were used to compare CUT&RUN results with the RNA-seq analysis. Gene ontology was performed for GO biological processes using ShinyGO (Ge et al., 2020) (https://doi.org/10.1093/bioinformatics/btz931), results were cutoff at FDR of 0.05 and the top 20 results were graphed. For neural stem cell CUT&RUN data, datasets were downloaded from (Pagin et al., 2021a) and processed as described above for the dLGN datasets. The two peak sets were overlapped with Bedtools intersect. STRING (Szklarczyk et al., 2023) was used to graph protein-protein interactions between the set of 79 genes that were found to be dLGN SOX2 targets coregulated by SOX2 and NR2F1, disconnected nodes were removed. Bedgraphs were visualized in IGV (Robinson et al., 2011).

## Accession to genomic data

RNA-seq data are accessible through Gene Expression Omnibus (GEO) Accession Number GSE233131. CUT&RUN data are accessible through ArrayExpress Accession Number E-MTAB-13004.

## Acknowledgements

We thank Dr D. O'Leary for providing the *RORα*-Cre mouse, Dr Carolina Frassoni and her laboratory for contributing to the IF experiments in Fig. S1B, and Dr Aixa Morales for supplying the anti-SOX5 antibody and *Sox5* ISH probe. The computations and data handling for CUT&RUN analyses were enabled by resources provided by the National Supercomputer Centre (NSC), funded by Linköping University. Peter Münger at the National Supercomputer Centre is acknowledged for assistance concerning technical and implementational aspects in making the codes run on the Sigma resource.

## Competing interests

The authors declare no competing or financial interests.

## Author contributions

Conceptualization: M.J., S.O., C.C., S.K.N., S.M.; Data curation: A.N., M.J., C.M., F.Z., G.P.; Formal analysis: L.S., A.N., M.J., F.G., F.Z., G.P., S.M.; Funding acquisition: M.S., C.C., S.K.N., S.M.; Investigation: L.S., M.J., C.M., G.R., A.D., F.G., F.Z., G.P.; Methodology: L.S., A.N., M.J., G.R., A.D., F.Z., G.P., C.C., S.M.; Project administration: M.S., C.C., S.K.N., S.M.; Resources: F.Z., G.P., C.C., S.K.N.; Software: G.P.; Supervision: M.J., M.S., G.P., C.C., S.K.N., S.M.; Validation: L.S., A.N., M.J., F.G., F.Z., G.R., A.D., S.M.; Visualization: L.S., A.N., M.J.; Writing – original draft: S.O., C.C., S.K.N.; Writing – review & editing: L.S., A.N., M.J., C.M., F.G., S.O., F.Z., M.S., G.P., C.C., S.K.N., S.M.

## Funding

Work in the Nicolis laboratory is supported by EU ERANET-NEURON grant Brain4Sight and by Fondazione Telethon – Fondazione Cariplo Alliance GJC21176

grant to S.K.N. Additional funding to S.M., G.R. and A.D. was provided by Fondazione Telethon grant GMR23T1022 and grant PRIN 2022 (Next Generation EU, Missione 4 Componente 1, CUP H53D23003020001) to S.M. M.S. was funded by the EU ERANET- NEURON grant Brain4Sight  (grant number: ANR- 21-NEU2-0003-03). L.S. was a recipient of a Vinci PhD Fellowship from the Universita' Italo-Francese. Work in the Cantù lab is supported by Cancerfonden (CAN 2018/542 and 21 1572 Pj), the Swedish Research Council, Vetenskapsrådet (2021–03075), Additional Ventures (USA) (SVRF2021-1048003), Linköping University support and LiU/RÖ-Cancer. C.C. is a Wallenberg Molecular Medicine (WCMM) fellow and receives generous financial support from the Knut and Alice Wallenberg Foundation. Open Access funding provided by Linköping University. Deposited in PMC for immediate release.

## Data and resource availability

All relevant data can be found within the article and its supplementary information.

## Peer review history

The peer review history is available online at https://journals.biologists.com/bio/lookup/doi/10.1242/bio.062014.reviewer-comments.pdf.

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
