## [Peer Review File · Biology Open]

SOX2 and NR2F1 coordinate the gene expression program of the early postnatal visual thalamus

Linda Serra, Anna Nordin, Mattias Jonasson, Carolina Marenco, Guido Rovelli, Annika Diebels, Francesca Gullo, Sergio Ottolenghi, Federico Zambelli, Michèle Studer, Giulio Pavesi and Claudio Cantù
DOI: 10.1242/bio.062014

Editor: Daniel Gorelick

Review timeline

Submission to Review Commons:	30 November 2023
Submission to Biology Open:	4 April 2025
Editorial decision:	10 April 2025
First revision received:	19 June 2025
Accepted:	2 July 2025

Reviewer 1:

Evidence, reproducibility and clarity

Serra et al have conducted transcriptomic analyses for thalamic Sox2 and Nr2f1 cKO mice, revealing gene regulatory networks underlying development and functions of dLGN which plays pivotal roles in visual sensation. The findings are also potentially important for understanding vision disability in human. Their conclusions are mostly supported by the data, but some reinforcement and additional explanations may further improve the paper.

****Major points:****

1. Although they showed that Sox2 does not regulate Nr2f1 by immunostaining in Fig.1, it would be reinforced by the RNA-seq results. What about evidence for regulation of Sox2 by Nr2f1? I could not find.
2. The onset of and specificity among the thalamic nuclei of Sox2 and Nr2f1 expression would better be mentioned in the beginning. As far as I remember, both genes are quite widely expressed in the thalamic nuclei, not necessarily specific to dLGN.
3. Mechanistically, how Sox2 function becomes distinct in neural stem cells and neurons would be of a great interest (e.g., changes in binding partner). But, it might be too much for the present package.

****Minor points:****

1. Explanation for the values in Fig.3A in the text or the figure legend would be helpful for readers unfamiliar with MuSiC.
2. Since Ror-alpha is also expressed layer 4 in the cortex, some explanations for these phenotypes being caused by thalamic defects may be provided. I know that expression of Sox2 and Ror-alpha do not overlap in layer 4, though.
3. Why did the authors use two types of Sox2 antibodies in Fig.4A?
4. Quantification for Fig.1A, Fig.2A and 2B may be necessary for the current publication standards.
5. In Introduction, NRF1 or NRF is somewhat confusing because there is a different gene named NRF (Nuclear respiratory factor).
6. Reference 14 is identical to 44.

Significance

This work provides a basis of gene regulatory network involved in development and function of dLGN neurons, which may also be important for understanding mechanisms of vision disability in human caused by genetic mutations.

Although I am not an expert in this particular field (GRNs in thalamic neurons), a series of the authors' works certainly establish a molecular basis of the roles of Sox2 ranging from neural stem/progenitor cells to neurons. Limitations of the current study in my opinion would be that it only lists up candidate genes for the functions or cause of visual sensations or defects, and thus experimental proof awaits actual biological experiments. Although the results and conclusion provided by the authors are reasonable and convincing, conceptual advance may be limited to some extent. Readers in both basic and clinical researches will be interested in that vision disability caused by mutations in Sox2 and Nr2f1 could be explained by synapse-related genes, axon guidance molecules, or secreting factors like VGF, albeit not with big surprise.

My research expertise would be in the field of brain development, particularly in regionalization and morphogenesis of the brain. Yet, I am not particularly familiar with transcriptomic analyses in general.

Reviewer 2:

Evidence, reproducibility and clarity

In the current manuscript, Serra, Mercurio, and colleagues carried out Ror-alpha-Cre specific conditional mutant analysis of Sox2 and Nr2f1 in the thalamus/dLGN. The workflow primarily focused on potential mechanisms underlying transcriptional regulation. With RNA-Seq, the authors identified multiple "common" targets shared by both Sox2 and Nr2f1 factors. In parallel, the authors also carried out CUT-RUN analysis for Sox2 binding patterns in dLGN chromatin.

The current work is built upon the intellectual framework of two papers: the past work led by the senior author in 2019, as well as an earlier work by Chou /O'Leary 2013, in terms of genetic reagents and anatomical and functional analysis. While the newly performed experiments may open some new avenues for future investigation, the current manuscript did NOT vigorously validate bioinformatics predictions using experimental approaches. The current dataset did NOT present any functional and anatomical analysis, esp. in terms of the target gene functions back to the same circuits/connections (thalamus-cortex).

The manuscript presented in the current format offers limited biological insights into the neurobiology of dLGN. The limited experimental data also indicated that the manuscript may not be suitable for a very general readership.

****Major points:****

1. Unless I missed anything - I was not sure why the current Figure 1/ Tables 1&2 took a sharp pause without any in situ/histochemical validations of the "prominent" downstream targets - at minimum, the authors should validate the common targets, including VGF among others;
2. Could the over-expression of any targets (Sox5, etc) reverse the loss of Sox2-phenotypes, esp. in terms of the establishment of thalamic-cortical connections, as assayed by Fig 2A (as well as Mercurio, 2019, Figure4)? Having such an assay would significantly boost the significance of the current study.
3. Figure 3 is presented in a very inconvenient manner for any reviewers/future readers to understand and interpret. The plots in B and C are what matter the most, while the raw data in 3A could be included in a table. The presentation and comparison of this figure need some significant work.
4. The Cut-n-Run assays offered several dLGN unique (non-neurogenesis) targets. However, the study paused at bioinformatics prediction without experimental validations as well, including the dLGN peaks near Vgf and Sox5.

****Minor points:****

For general readers, (1) please explicitly document whether Ror-alpha-Cre does NOT(?) impact the retina and cortex; (2) please explain when Ror-alpha-Cre expression timing - is it solely post-mitotic in the dLGN? The authors may have taken these for granted, esp. given Mercurio 2019 and Chou 2013, but such information may help readers outside the field.

Significance

The manuscript offers limited new information to general readers. It might be a good dataset for researchers specialized in transcriptional regulation in terms of finding useful/relevant information to design future experiments. However, the study did NOT offer any histological and functional assays based on bioinformatics tests.

General assessment:

The strengths were a careful analysis of dLGN in early development using both RNA-Seq and Cut-n-Run with a focus on Sox2's post-mitotic role. The limitations were that the study was lack of histological validations and functional tests of the candidate genes.

Advance:

The advance of the study is limited, though the experiments were carefully launched.

Audience:

Very limited audience with a specialty in transcription factors in visual system development. The reviewer is an expert in neurodevelopment using the mouse genetics approach, with primary interests in studying the retina and retino-recipient zone development.

Reviewer 3:

Evidence, reproducibility and clarity

Summary:

This manuscript investigates the role of Sox2 and Nr2f1 on dLGN development. The authors perform RNA-seq on thalamus-specific conditional knock outs of Sox2 and Nr2f1. The author compile lists of the genes that showed the greatest change in detection between control mice (3 and 3) and mutant mice (3 and 3). The authors find significant overlap in the lists of genes most altered in the mutants and argue that this overlap is consistent with the two transcription factors regulating the same gene network. The authors also perform a CUT&RUN analysis of Sox2 binding sites and find overlap in the list of genes that Sox2 binds to and the genes with altered expression levels in the Sox2-ckO. Regulation of neuron-specific cellular components are highly represented in both the list of binding sites and genes with altered expression levels.

The RNA-seq data and binding site data are valuable resources for researchers trying to understand the development of the dLGN and should be published. However, I am not confident that author's interpretations of their data are supported by what is provided in the manuscript.

Major comments:

Issues with the statistical logic

- Lack of statistical significance is not evidence of equality. The fact that Sox2 and Nr2f1 do not pass the FDR threshold is not evidence that they are unchanged in the two conditional knockouts.

- Many arguments are based on the result that Sox2 knock out has a "strong" effect on a gene. FDR and p-values do not provide evidence about effect size beyond "not 0". Average TPN values are provided but, without sorting through thousands of values in the supplementary data, it is not possible to judge the reliability of a claimed effect size. Finally, no biological reference is given for what should be considered a strong effect size besides the relative values within the

knockout experiment. I would like to see the replicates for the relevant TPN data presented in the main text and I would like to see the variance between those replicates considered in the author's conclusions. Space in the tables could be saved by reporting fewer digits in the fold changes.

- The authors identify 469 dLGN specific SOX2 binding sites by subtracting the 248 high confidence binding sites identified in non-dLGN cells from the 717 high confidence binding sites identified in dLGN. This subtraction is basically a comparison of p-values with the false assumption that lack of statistical significance means there was no change. The quantitation required to make the claim would be a direct comparison of the two data sets for each binding site.

Non-quantitative issues:

- It is known that both the Sox2 and Nr2f1 mutants have similar dLGN phenotypes. How, then, can we know if individual changes in gene expression reflect direct regulation by Sox2 and Nr2f1 or the dramatically altered state of the dLGN? The binding data would add to the argument of direct regulation, but it is difficult to judge the specificity of the binding data.
- The authors argue that a decrease in layer 4 of the cortex argues that Vgf1 is a likely link between Sox2 and cortical development. However, some decrease in layer 4 thickness is a given if the number of thalamocortical cells in dLGN is reduced.
- Immuno fluorescence is used to support the idea that the number of cells strongly expressing Sox5 is reduced in the Sox2 cKO. The image shows a reduced patch of Sox5 labeling. However, the dLGN is generally reduced in the Sox2 cKO so it is not clear if there is a difference in the proportion of cells expressing Sox5. The sample size also appears to be 1.

Minor

Introduction:

- Writing could be improved.
- Descriptions of effects of Sox2 or Nr2f1 using RORalpha-Cre use words like "reduced", "significant", "important". It is unclear what the actual effects or effect sizes are.

RESULTS

- What is "Three independent pools of mutant and control dissected visual thalami"? Three mice for each condition (twice for control)?
- Why are there two groups of 3 control mice each and not one group of 6? Section 2

Section 2

- For the model in which the probability of genes changing in the same direction is calculated, are all genes assumed to have the same chance of passing the FDR? Gene variance and detection rate will be correlated between conditions. I would suggest a more conservative comparison. What is the correlation of fold change for genes that pass FDR? Of 514 that change in both, 481 go in the same direction and 33 go in a different direction. If everything is random, the number would be 257/257. The claim of four times random overlap does not seem like the conservative estimate.

Section 3

- I don't see any basis to judge the p-values in Fig 1D. How do these changes compare to what you would from other dramatic manipulations of neural tissue? Can figure 1D compare to changes in non-neuronal standard? How about metabolism and cell death?

Section "Deconvolution..."

- It is great that results for each replicate is presented.
- There are too many significant digits in Fig 3A given the variance.

- Why do the NR2F1 mutants look more like the Sox2 controls (in terms of excitatory Neurons) than the NR2F1 controls do?

Section "CUT&RUN..."

- How many overlaps (Figure 4B) would you expect by chance?
- Fig 4J needs more description. What does the first full pie represent?
- Please include the denominator in the binding event argument. It is difficult to judge the specificity of the effect in this section.

Significance

The mouse dorsal lateral geniculate nucleus (dLGN) is an important model system for understanding vision and the development of visual circuitry. A considerable literature exists on the role of activity dependent development and molecular gradients in shaping the synaptic connections between the retina and the dLGN. Less is known about the transcriptional networks that regulate dLGN development. Mutations in the transcription factors Sox2 and NR2F1 are associated with severe vision defects and conditional knockout of Sox2 has been shown to cause dramatic defects in dLGN development. The data provided in the current study adds to our understanding of how these transcription factors influence gene expression and circuit formation in the dLGN. Their work points to changes in VGF expression and fewer thalamocortical cells as the most salient effects of Sox2 deletion. These results increase our understanding of the transcriptional networks underlying dLGN development and several visual pathologies.

I think the manuscript should be helpful to researchers interested in the dLGN or researchers interested in the transcription factors important for neural circuit development. My own expertise covers dLGN development but not transcription factors and the interpretation of RNA-seq data. My impression was that the biggest contribution of this manuscript was in obtaining gene expression levels in the Sox2 conditional knockout with multiple RNA-seq replicates. The impact of the paper, as written, is lessened by the fact that the confidence gained by replicating the analysis is not leveraged in the main text of the manuscript. Much of the results, interpretation, and discussion depend on sorting strong effects on genes from weak ones without presenting replicates for effect size or confidence intervals. The replicate data is available in the supplementary data and should be a good resource for future research.

First revision

Author responses to Reviewer comments

Reviewer #1 (Evidence, reproducibility and clarity (Required)):

Serra et al have conducted transcriptomic analyses for thalamic Sox2 and Nr2f1 cKO mice, revealing gene regulatory networks underlying development and functions of dLGN which plays pivotal roles in visual sensation. The findings are also potentially important for understanding vision disability in human. Their conclusions are mostly supported by the data, but some reinforcement and additional explanations may further improve the paper.

We thank the reviewer for their appreciation of our work, and the constructive comments.

Major points:

1. Although they showed that Sox2 does not regulate Nr2f1 by immunostaining in Fig.1, it would be reinforced by the RNA-seq results. What about evidence for regulation of Sox2 by Nr2f1? I could not find.

We have now highlighted, in Fig. 1D, the requested RNAseq results from Table S1, showing a very limited reduction of expression of Nr2f1 in Sox2 mutant and of Sox2 in Nr2f1 mutants. We further added ISH results confirming this data (Fig. 4A).

2. The onset of and specificity among the thalamic nuclei of Sox2 and Nr2f1 expression would better be mentioned in the beginning. As far as I remember, both genes are quite widely expressed in the thalamic nuclei, not necessarily specific to dLGN.

We previously reported in Mercurio et al 2019 (ref. 7) that Sox2 is highly expressed in the dorsal thalamus (precursor to the sensory thalamic nuclei) at least from E15.5 and is later expressed in all the sensory thalamic nuclei, though not in surrounding regions (Mercurio et al 2019 Fig. 1). A similar expression pattern was previously reported for Nr2f1 in Chou et al 2013 (ref. 6). A brief mention of this point is now present in Introduction.

3. Mechanistically, how Sox2 function becomes distinct in neural stem cells and neurons would be of a great interest (e.g., changes in binding partner). But, it might be too much for the present package.

We agree on the interest of this point. We note that SOX2 binding sites in neurons (but not in stem cells), as detected by CUT&RUN, are enriched for SOX2 and RORA/NRF binding sites. The co-presence of SOX and NRF potential binding motifs (Fig. 2F-G), suggests the possibility of direct physical interaction between SOX2 and NR2F1 mediating joint binding to DNA. This is interesting and will be experimentally addressed in a follow up study.

Minor points:

1. Explanation for the values in Fig. 3A in the text or the figure legend would be helpful for readers unfamiliar with MuSiC.

We clarified the figure legend, better explaining how the plotted were computed and their meaning.

2. Since Ror-alpha is also expressed layer 4 in the cortex, some explanations for these phenotypes being caused by thalamic defects may be provided. I know that expression of Sox2 and Ror-alpha do not overlap in layer 4, though.

In fact, we propose that downregulation of RORa in layer 4 maybe caused by reduced thalamic afferents to layer 4, possibly also acting through a reduced delivery of VGF to the cortex; in fact, as the reviewer correctly states Sox2 itself is not expressed in the cortex.

3. Why did the authors use two types of Sox2 antibodies in Fig. 4A?

We strive to replicate our CUT&RUN data such that we can rely only on the reproducible binding events. We have often noted that - being CUT&RUN a “challenging” application for antibodies - different antibodies yield non-fully overlapping binding profiles. While we do not have a clear explanation for this, we consider more robust converging on those binding events that are obtained by two independent antibodies, when such tools are available. This, in our opinion and experience, drastically decreases the chance of stumbling upon false positive hits.

4. Quantification for Fig. 1A, Fig. 2A and 2B may be necessary for the current publication standards.

The requested quantification has been added in Fig. S1A and in Fig. 4C.

5. In Introduction, NRF1 or NRF is somewhat confusing because there is a different gene named NRF (Nuclear respiratory factor).

We corrected this.

6. Reference 14 is identical to 44.

We corrected this.

Reviewer #1 (Significance (Required)):

This work provides a basis of gene regulatory network involved in development and function of dLGN neurons, which may also be important for understanding mechanisms of vision disability in human caused by genetic mutations.

Although I am not an expert in this particular field (GRNs in thalamic neurons), a series of the authors' works certainly establish a molecular basis of the roles of Sox2 ranging from neural stem/progenitor cells to neurons. Limitations of the current study in my opinion would be that it only lists up candidate genes for the functions or cause of visual sensations or defects, and thus experimental proof awaits actual biological experiments. Although the results and conclusion provided by the authors are reasonable and convincing, conceptual advance may be limited to some extent. Readers in both basic and clinical researches will be interested in that vision disability caused by mutations in Sox2 and Nr2f1 could be explained by synapse-related genes, axon guidance molecules, or secreting factors like VGF, albeit not with big surprise.

My research expertise would be in the field of brain development, particularly in regionalization and morphogenesis of the brain. Yet, I am not particularly familiar with transcriptomic analyses in general.

Reviewer #2 (Evidence, reproducibility and clarity (Required)):

In the current manuscript, Serra, Mercurio, and colleagues carried out Ror-alpha-Cre specific conditional mutant analysis of Sox2 and Nr2f1 in the thalamus/dLGN. The workflow primarily focused on potential mechanisms underlying transcriptional regulation. With RNA-Seq, the authors identified multiple "common" targets shared by both Sox2 and Nr2f1 factors. In parallel, the authors also carried out CUT-RUN analysis for Sox2 binding patterns in dLGN chromatin.

The current work is built upon the intellectual framework of two papers: the past work led by the senior author in 2019, as well as an earlier work by Chou /O'Leary 2013, in terms of genetic reagents and anatomical and functional analysis. While the newly performed experiments may open some new avenues for future investigation, the current manuscript did NOT vigorously validate bioinformatics predictions using experimental approaches. The current dataset did NOT present any functional and anatomical analysis, esp. in terms of the target gene functions back to the same circuits/connections (thalamus-cortex).

The manuscript presented in the current format offers limited biological insights into the neurobiology of dLGN. The limited experimental data also indicated that the manuscript may not be suitable for a very general readership.

We thank the reviewer for pointing out contributions as well as limitations of our work. We are convinced that our work does indeed open up "new avenues for future investigation", reporting for the first time hundreds of targets of SOX2 and NR2F1 as well as hundreds of direct SOX2 binding sites in dLGN neurons that will contribute to future investigations.

Major points:

1. Unless I missed anything - I was not sure why the current Figure 1/ Tables 1&2 took a sharp pause without any in situ/histochemical validations of the "prominent" downstream targets - at minimum, the authors should validate the common targets, including VGF among others;

We now validated the downregulation VGF and Sox5 at the RNA level by ISH confirming SOX5 downregulation by IF. These data are presented in the new Fig. 4, in results page 5 and discussion page 7.

2. Could the over-expression of any targets (Sox5, etc) reverse the loss of Sox2-phenotypes, esp. in terms of the establishment of thalamic-cortical connections, as assayed by Fig 2A (as well as Mercurio, 2019, Figure4)? Having such an assay would significantly boost the significance of the current study.

The experiment suggested by the reviewer would undoubtedly be interesting to address Sox5 contribution to the mutant phenotype; unfortunately, this is too demanding for the present paper.

However, for the sake of data interpretation, we propose that the mutant phenotypes observed rather result from the global deregulation of a set of genes, not just of a single gene. Indeed, we discuss the potential contribution of several different genes, among those co-regulated by SOX2 and NR2F1. From this point of view, we don't necessarily expect the contribution of a specific gene to be prominent. In fact, we believe an interesting result emerging from our work is the identification of a rather numerous set of genes collectively responding to both Sox2 and Nr2f1 mutation, many of which may contribute to the shared phenotypes of the two mutants.

3. Figure 3 is presented in a very inconvenient manner for any reviewers/future readers to understand and interpret. The plots in B and C are what matter the most, while the raw data in 3A could be included in a table. The presentation and comparison of this figure need some significant work.

We have now modified Fig. 3 as requested and moved the raw data to the Supplementary material (Table S4).

4. The Cut-n-Run assays offered several dLGN unique (non-neurogenesis) targets. However, the study paused at bioinformatics prediction without experimental validations as well, including the dLGN peaks near Vgf and Sox5.

We are not sure we understand the reviewer's question. The "dLGN unique (non-neurogenesis) targets" that we report are not the results of a bioinformatics prediction, but of the CUT&RUN experiment itself including the dLGN peaks near Vgf and Sox5. In addition, we experimentally validated the downregulation of Vgf and Sox5 by in situ hybridization in the new Figure 4.

Minor points:

For general readers, (1) please explicitly document whether Ror-alpha-Cre does NOT(?) impact the retina and cortex;

This is now mentioned in results in agreement with the results in Chou et al. 2013 and Mercurio et al. 2019.

Chou et al mentions explicitly absence of Rora Cre activity in the cortex and this is also in agreement with our own results in Mercurio et al. 2019. As to the retina, we reported not observing any retinal phenotypes in Sox2 mutants in agreement with the absence of any Sox2 deletion within the retina, that would have caused a drastic phenotype as reported in Taranova et al. 2006.

(2) please explain when Ror-alpha-Cre expression timing - is it solely post-mitotic in the dLGN? The authors may have taken these for granted, esp. given Mercurio 2019 and Chou 2013, but such information may help readers outside the field.

The onset of Rora Cre activity is at a stage in which dLGN neurogenesis is completed and most if not all cells are postmitotic as reported in Chou et al. 2013. This point is now more explicitly mentioned in results.

Reviewer #2 (Significance (Required)):

The manuscript offers limited new information to general readers. It might be a good dataset for researchers specialized in transcriptional regulation in terms of finding useful/relevant information to design future experiments. However, the study did NOT offer any histological and functional assays based on bioinformatics tests.

- General assessment:

The strengths were a careful analysis of dLGN in early development using both RNA-Seq and Cut-n-Run with a focus on Sox2's post-mitotic role. The limitations were that the study was lack of histological validations and functional tests of the candidate genes.

We now added histological validation of selected targets as requested in the new Fig. 4.

- Advance:

The advance of the study is limited, though the experiments were carefully launched.

- Audience:

Very limited audience with a specialty in transcription factors in visual system development.

The reviewer is an expert in neurodevelopment using the mouse genetics approach, with primary interests in studying the retina and retino-recipient zone development.

Reviewer #3 (Evidence, reproducibility and clarity (Required)):

Summary:

This manuscript investigates the role of Sox2 and Nr2f1 on dLGN development. The authors perform RNA-seq on thalamus-specific conditional knock outs of Sox2 and Nr2f1. The author compile lists of the genes that showed the greatest change in detection between control mice (3 and 3) and mutant mice (3 and 3). The authors find significant overlap in the lists of genes most altered in the mutants and argue that this overlap is consistent with the two transcription factors regulating the same gene network. The authors also perform a CUT&RUN analysis of Sox2 binding sites and find overlap in the list of genes that Sox2 binds to and the genes with altered expression levels in the Sox2-cKO. Regulation of neuron-specific cellular components are highly represented in both the list of binding sites and genes with altered expression levels.

The RNA-seq data and binding site data are valuable resources for researchers trying to understand the development of the dLGN and should be published. However, I am not confident that author's interpretations of their data are supported by what is provided in the manuscript.

Major comments:

Issues with the statistical logic

-Lack of statistical significance is not evidence of equality. The fact that Sox2 and Nr2f1 do not pass the FDR threshold is not evidence that they are unchanged in the two conditional knock-outs.

The meaning of statistical testing and significance in this context is assessing if, and how much, the observed changes in expression in RNA-Seq estimated transcript levels can be due only to experimental variability (not significant) or, vice versa, if there is an additional biological factor (the knock-out of Sox2 or Nr2f1, in this case) behind the changes observed. Clearly, the more "significant" (lower) are the p-value/FDR values associated with changes observed for a gene, the more likely is that the gene transcript levels are affected by the knock outs. Vice versa, if the change is reported to be "not significant", there isn't enough evidence - at least from a statistical point of view - that the observed changes in transcript levels are due to the effect of the knock outs. Three replicates per condition are required in order to estimate variance - which is gene specific and estimates what is the "natural" range of variability of each gene due only to experimental variability (and not generated by the knock-outs).

We now report the RNAseq data for Sox2 and Nr2f1 in Fig. 1D and complete them with ISH data in the new Fig. 4. The results are consistent with a limited reduction Nr2f1 in the Sox2 mutants and Sox2 in the Nr2f1 mutants. Though we cannot rule out that they might contribute to some extent to the mutant phenotype, we document a stronger downregulation, in both mutants, of a vast set of other genes (Fig. 1C) onto which our analysis focuses.

-Many arguments are based on the result that Sox2 knock out has a "strong" effect on a gene. FDR and p-values do not provide evidence about effect size beyond "not 0". Average TPN values are provided but, without sorting through thousands of values in the supplementary data, it is not possible to judge the reliability of a claimed effect size. Finally, no biological reference is given for what should be considered a strong effect size besides the relative values within the knockout experiment. I would like to see the replicates for the relevant TPN data presented in the main text and I would like to see the variance between those replicates considered in the author's conclusions. Space in the tables could be saved by reporting fewer digits in the fold changes.

See previous point. The more “significant” are the changes of transcript levels according to statistical testing, the “stronger” the effect of the knock out on them, where by “strong” we mean a more relevant variation of transcript levels. However, since we realized that this term could cause confusion in the reader, we rephrased the relevant parts. Variance is taken into account in the computation of p-values/FDRs, so the same difference in mean TPM values for two different genes can result to more/less significant according to the estimated variance of the values.

-The authors identify 469 dLGN specific SOX2 binding sites by subtracting the 248 high confidence binding sites identified in non-dLGN cells from the 717 high confidence binding sites identified in dLGN. This subtraction is basically a comparison of p-values with the false assumption that lack of statistical significance means there was no change. The quantitation required to make the claim would be a direct comparison of the two data sets for each binding site.

We appreciate the concern from the reviewer. CUT&RUN, especially when performed in vivo versus cell lines, has a high intrinsic variability between experiments, and even between technical replicates (DOI: 10.1093/nar/gkx180). While it would be possible to, for example, run DiffBind (built for ChIP-seq), on the dLGN data versus the NS data, these are not, in our opinion, directly comparable as they were not performed in the same batch, on the same type of material (dissected mouse tissue versus cultured cells) or even with the same batches of reagents. Thus, to quantify them in terms of signal at specific loci, without taking into account things like global background, local background, and overall signal to noise ratio, we do not believe is correct. There are many attempts in the field to better quantify CUT&RUN data (spike-in yeast or E. coli DNA at different moments, spike-in drosophila nuclei, etc.) but there remains to be determined a general consensus on what is best or trustworthy. The best way we could do the comparison, with our data as it was generated, was as pointed out above, by comparing the statistically significant events in the dLGN versus those in the NS, that way each dataset is considered independently before the overlap is performed. To help alleviate the reviewers concerns, we have provided here, for the reviewer, signal profiles and heatmaps of the dLGN only regions in both dLGN and NS CUT&RUN.

Non-quantitative issues:

-It is known that both the Sox2 and Nr2f1 mutants have similar dLGN phenotypes. How, then, can we know if individual changes in gene expression reflect direct regulation by Sox2 and Nr2f1 or the dramatically altered state of the dLGN? The binding data would add to the argument of direct regulation, but it is difficult to judge the specificity of the binding data.

The timepoint of the RNAseq analyses was chosen to precede any phenotypic changes detected in the dLGN based on our previous analyses reported in Mercurio et al. 2019 as stated in Results page 3.

-The authors argue that a decrease in layer 4 of the cortex argues that Vgf1 is a likely link between Sox2 and cortical development. However, some decrease in layer 4 thickness is a given if the number of thalamocortical cells in dLGN is reduced.

We agree with the Reviewer. The possible contribution of VGF has been rephrased considering a possible wider contribution of thalamic afferents in general.

-Immuno fluorescence is used to support the idea that the number of cells strongly expressing Sox5 is reduced in the Sox2 cKO. The image shows a reduced patch of Sox5 labeling. However, the dLGN is generally reduced in the Sox2 cKO so it is not clear if there is a difference in the proportion of cells expressing Sox5. The sample size also appears to be 1.

The time of this analysis was chosen to precede dLGN size reduction in mutants, as clearly shown in our previous work Mercurio et al. 2019 and further confirmed by the new ISH for Sox2 and Nr2f1 presented in the new Fig. 4.

The sample size is n=4 as reported in the Figure legend.

Minor

Introduction:

-Writing could be improved.

-Descriptions of effects of Sox2 or Nr2f1 using RORalpha-Cre use words like "reduced", "significant", "important". It is unclear what the actual effects or effect sizes are.

We revised the wording for this point.

RESULTS

-What is "Three independent pools of mutant and control dissected visual thalami"? Three mice for each condition (twice for control)?

-Why are there two groups of 3 control mice each and not one group of 6?

As reported in Materials and Methods " RNA sequencing was performed on three independent samples for both mutant and control dLGN. Each sample was composed of dLGNs from three animals of the same genotype pooled together."

Thalami from 3 mice represent an adequate amount of RNA to perform a single experiment of RNAseq. 3 x 3 represents a biological triplicate for the RNAseq experiment.

Section 2

-For the model in which the probability of genes changing in the same direction is calculated, are all genes assumed to have the same chance of passing the FDR? Gene variance and detection rate will be correlated between conditions. I would suggest a more conservative comparison. What is the correlation of fold change for genes that pass FDR? Of 514 that change in both, 481 go in the same direction and 33 go in a different direction. If everything is random, the number would be 257/257. The claim of four times random overlap does not seem like the conservative estimate.

Genes were selected with the same FDR thresholds in both experiments. The assumption is anyway more simple: the probability of a gene to have a significant change (passing the FDR threshold) in one experiment does not influence its probability to change also in the other, and vice versa. That is, we compute the probability to have a given number of up- or down-regulated genes in common in the two experiments assuming that the two experiments were independent from one another. From another point of view, this is the usual strategy employed in order to assess whether the overlap between two gene sets obtained by two different genome-wide experiments can be considered to be random or not, that is, if the number of genes in the overlap is close to random expected values they can be considered to be independent from one another.

Section 3

-I don't see any basis to judge the p-values in Fig 1D. How do these changes compare to what you would from other dramatic manipulations of neural tissue?
Can figure 1D compare to changes in non-neuronal standard? How about metabolism and cell death?

The graph shown represents the most significantly enriched functional annotations (GO annotations, pathways, etc.) among the deregulated genes as computed by Enrichr, one of the many tools developed for this task. And as for all the tools performing this analysis, the p-value means "the probability of having the same number of genes sharing the same functional annotation in a set of genes chosen at random", computed with the same strategy employed for the overlap between the two deregulated gene sets described before.

Section "Deconvolution..."

-It is great that results for each replicate is presented.

We thank the reviewer.

-There are too many significant digits in Fig 3A given the variance.

This has been adjusted as suggested.

-Why do the NR2F1 mutants look more like the Sox2 controls (in terms of excitatory Neurons) than the NR2F1 controls do?

The graphical presentation of the data in Fig. 3 has been improved, and the numerical data (former panel A) have been moved to the supplementary materials (Table S4) as recommended. Controls for Nr2f1 and Sox2 mutants have similar values for excitatory neurons, as expected, see Table S4. Fig. 3 shows the variation between each knock-out and its respective control experiments, and although excitatory neurons are reduced in both mutants the extent of reduction is greater in the Sox2 mutant.

Section "CUT&RUN..."

-How many overlaps (Figure 4B) would you expect by chance?

This is an extremely difficult number to calculate. It is possible to, for example, generate a random set of genomic fragments of similar length, and check how many of them overlap. This would however be extremely unfair, as CUT&RUN is naturally biased towards open chromatin, and thus would preferentially contain these types of regions in a "randomly" digested set. Additionally, data analysis and mapping biases further increase what overlaps would often occur. To circumvent this, we i) use an IgG control, which should identify and remove regions that are nonspecifically digested and sequenced during the experiment, and ii) performed our analysis after first removing sets of known artifact regions (Nordin et al 2023, ref. 43).

-Fig 4J needs more description. What does the first full pie represent?

*We have added more description in the figure legend, it now reads:
J. Schematic depiction of CUT&RUN and RNA-seq overlap, showing Sox2 peak associated genes that are transcribed (> 5 TPM, 784/1102) and those that are differentially expressed (DEG) in Sox2 mutant dLGN (FDR < 0.01, 327/784), and those that are up- (92) or down- (145) regulated in the mutant.*

-Please include the denominator in the binding event argument. It is difficult to judge the specificity of the effect in this section.

We apologize but we don't understand this comment.

Reviewer #3 (Significance (Required)):

The mouse dorsal lateral geniculate nucleus (dLGN) is an important model system for understanding vision and the development of visual circuitry. A considerable literature exists on the role of activity dependent development and molecular gradients in shaping the synaptic connections between the retina and the dLGN. Less is known about the transcriptional networks that regulate dLGN development. Mutations in the transcription factors Sox2 and NR2F1 are associated with severe vision defects and conditional knockout of Sox2 has been shown to cause dramatic defects in dLGN development. The data provided in the current study adds to our understanding of how these transcription factors influence gene expression and circuit formation in the dLGN. Their work points to changes in VGF expression and fewer thalamocortical cells as the most salient effects of Sox2 deletion. These results increase our understanding of the transcriptional networks underlying dLGN development and several visual pathologies.

I think the manuscript should be helpful to researchers interested in the dLGN or researchers interested in the transcription factors important for neural circuit development. My own expertise covers dLGN development but not transcription factors and the interpretation of RNA-seq data. My impression was that the biggest contribution of this manuscript was in obtaining gene expression levels in the Sox2 conditional knockout with multiple RNA-seq replicates. The impact of the paper, as written, is lessened by the fact that the confidence gained by replicating the analysis is not leveraged in the main text of the manuscript.

Performing a RNA-Seq analysis in replicates is common practice, and as we detailed in our replies to the reviewer's comments the goal of replicates is to have reliable estimations of the parameters needed (mean, variance of each gene) for the subsequent statistical analyses. So, we leveraged the information obtained from the replicates in order to identify with high confidence with genes could be considered to be affected by the knock-outs.

Much of the results, interpretation, and discussion depend on sorting strong effects on genes from weak ones without presenting replicates for effect size or confidence intervals. The replicate data is available in the supplementary data and should be a good resource for future research.

As discussed in the previous responses, the statistical evaluations usually performed on estimated transcript levels and their variance can be translated into a more qualitative evaluation of the effect of the knock-outs performed - the larger is the impact on transcript levels of a gene with respect to its estimated variance (variability) the stronger the effect is assumed to be. Confidence intervals are not usually employed in this context - the "confidence" with which the experimental setting can be assumed to affect gene expression is summarized by the p-values and the subsequent FDR values.

Original submission

First decision letter

MS ID#: bio.062014

MS Title: SOX2 and NR2F1 coordinate the gene expression program of the early postnatal visual thalamus

Authors: Linda Serra, Anna Nordin, Mattias Jonasson, Carolina Marengo, Guido Rovelli, Annika Diebels, Francesca Gullo, Sergio Ottolenghi, Federico Zambelli, Michèle Studer, Giulio Pavesi and Claudio Cantù

Many thanks for transferring your manuscript to Biology Open from Review Commons. I have now had the chance to review your documents, and I'm pleased to say we'd love to publish your manuscript, pending a few small and minor changes that should not require any additional rounds of peer review. I very much appreciate the way you revised your manuscript and addressed the reviewer comments. Below are the small changes, consistent with Biology Open's policies on transparency and accessibility, requested for publication:

- all figures should be color blind friendly (eg IF images in Fig 1A should not use shades of red and green together)

- all graphs should show individual data points or be formatted as superplots (eg Fig 4C, <https://pmc.ncbi.nlm.nih.gov/articles/PMC7265319/>)

- please include the signal profiles and heatmaps of the dLGN only regions in both dLGN and NS CUT&RUN as a supplementary figure. This data was included in the response to reviewer

comments, but it should also be integrated into and included in the manuscript. It would be a shame to save this beautiful supporting data for just the reviewers and not share it with the world!

Thank you again for using Review Commons and for choosing to submit to The Company of Biologists journals, we are grateful for your participation and appreciate the chance to publish your manuscript.

Second decision letter

MS ID#: bio.062014R1

MS Title: SOX2 and NR2F1 coordinate the gene expression program of the early postnatal visual thalamus

Authors: Linda Serra, Anna Nordin, Mattias Jonasson, Carolina Marengo, Guido Rovelli, Annika Diebels, Francesca Gullo, Sergio Ottolenghi, Federico Zambelli, Michèle Studer, Giulio Pavesi and Claudio Cantù

Thank you for giving Biology Open an opportunity to assess your manuscript following submission to Review Commons. I am happy to tell you that your manuscript has been accepted for publication in Biology Open, pending our standard publication integrity checks. It was accepted on 02 July 2025.